# Sub-threshold neuronal activity and the dynamical regime of cerebral cortex

Oren Amsalem [1], Hidehiko Inagaki [2], Jianing Yu[3], Karel Svoboda [4] & Ran Darshan [5,6,7,8] ✉

Cortical neurons exhibit temporally irregular spiking patterns and heterogeneous firing rates. These features arise in model circuits operating in a 'fluctuation-driven regime', in which fluctuations in membrane potentials emerge from the network dynamics. However, it is still debated whether the cortex operates in such a regime. We evaluated the fluctuation-driven hypothesis by analyzing spiking and sub-threshold membrane potentials of neurons in the frontal cortex of mice performing a decision-making task. We showed that while standard fluctuation-driven models successfully account for spiking statistics, they fall short in capturing the heterogeneity in sub-threshold activity. This limitation is an inevitable outcome of bombarding single-compartment neurons with a large number of pre-synaptic inputs, thereby clamping the voltage of all neurons to more or less the same average voltage. To address this, we effectively incorporated dendritic morphology into the standard models. Inclusion of dendritic morphology in the neuronal models increased neuronal selectivity and reduced error trials, suggesting a functional role for dendrites during decision-making. Our work suggests that, during decision-making, cortical neurons in high-order cortical areas operate in a fluctuation-driven regime.

Cortical neurons spike at irregular times, with statistics that can be approximated by a Poisson process[1–3]. In addition, neuronal firing rates are highly heterogeneous, with some that fire at high rates while a large number of neurons are almost quiescent[4–7]. Ongoing research in system neuroscience is directed at understanding this operating regime of cortex[8], in which spiking irregularity and rate heterogeneity are ubiquitous.

Sensory input and movement can contribute to irregular spiking and heterogeneous spike rates[9,10]. However, substantial contribution to the variability and heterogeneity likely results from the recurrent network dynamics. Indeed, theoretical studies have shown that these features can emerge from the non-linear recurrent dynamics of cortical networks, if they operate in a 'fluctuation-driven regime' in which

excitatory and inhibitory currents are both large compared to the rheobase of the neurons but approximately balanced[11–14]. The net input can be sub-threshold, on average, while action potentials are driven by fluctuations in synaptic inputs[15].

Theoretical studies of the fluctuation-driven regime are foundational to our quantitative understanding of spiking statistics in the cortex[16–22]. However, it is still debated whether the cortex operates in such a regime[8]. For example, fluctuation-driven networks can quantitatively account for feature selectivity in the cortex[23–25], and are consistent with the observed balance of excitation and inhibition in neural recordings[26,27], as well as with cortical responses to perturbations in sensory and frontal areas[28–30]. On the other hand, similar perturbation experiments in layer 4 neurons of the barrel cortex are inconsistent

[1]Division of Endocrinology, Diabetes and Metabolism, Beth Israel Deaconess Medical Center, Harvard Medical School, Boston, MA, USA. [2]Max Planck Florida Institute for Neuroscience, Jupiter, FL, USA. [3]School of Life Sciences, Peking University, Beijing, China. [4]Allen Institute for Neural Dynamics, Seattle, WA, USA. [5]Department of Physiology and Pharmacology, Faculty of Medicine and Health Sciences, Tel Aviv University, Tel Aviv, Israel. [6]The School of Physics and Astronomy, Tel Aviv University, Tel Aviv, Israel. [7]The Sagol School of Neuroscience, Tel Aviv University, Tel Aviv, Israel. [8]Janelia Research Campus, Howard Hughes Medical Institute, Ashburn, VA, USA. ✉e-mail: darshan@tauex.tau.ac.il

with a fluctuation-driven hypothesis[31]. Furthermore, while some modeling studies suggest that recurrent and external inputs to cortical neurons are large[23,28,29], it has been argued that feedforward thalamic inputs to the visual cortex are too weak to be approximately balanced by a strong recurrent inhibition[8].

Standard models of networks that operate in the fluctuation-driven regime, in which synaptic interactions are mediated by variations in the synaptic conductance, predict that neuronal membrane potential should hover close to the neuronal threshold[32] and, as we will show, with limited heterogeneity in mean voltage across neurons. In this study, we tested the fluctuation-driven regime hypothesis by analyzing the supra- and sub-threshold activity of populations of excitatory and inhibitory neurons in the anterior lateral motor cortex (ALM) in behaving mice that perform a decision-making task. Does the same mechanism for explaining spiking statistics in cortex accounts for the variability and heterogeneous sub-threshold voltage activity?

We demonstrate that fluctuation-driven networks can account for spiking statistics in ALM. However, these models fail to reproduce the large level of voltage heterogeneity observed in the data. We show that this limitation arises from the strong excitatory and inhibitory synaptic inputs, leading to a significant increase in the neuronal input conductance. We resolve this discrepancy by introducing a phenomenological network model of 'extended-like' point neurons with synapses that mimic dendritic morphology, thus including a key aspect of neuronal biophysics that is often neglected in models of recurrent neural networks[33]. Our proposed model suggests that neurons in ALM are fluctuation-driven. Furthermore, effectively inclusion of dendrites in the model enhances neuronal selectivity during preparatory activity and reduces error trials, indicating a functional role for dendritic integration in decision-making.

## Results

### Spiking and sub-threshold activity in ALM

We analyzed the supra (spiking) threshold activity of excitatory and inhibitory cells and the sub-threshold activity (membrane potential) of excitatory cells of the anterior lateral motor cortex (ALM) of mice performing a delayed-response task (analysis of data from[34,35]). In each trial, mice were instructed to lick to one of two lick ports (sample period). During the subsequent delay epoch, mice maintained a memory of the previous sensory experience and planned an upcoming response. Following an auditory 'go' cue, mice reported the sensory instruction by directional licking (left or right). (Fig. 1A,[36]). We analyzed two types of data sets, in which the instruction cue was either auditory ('auditory task'; silicon probes or whole-cell recordings) or tactile ('tactile task'; whole-cell recordings). See[35] and Methods for further details). In ALM a large proportion of recorded neurons exhibited preparatory activity that predicts licking direction[35].

Consistent with previous recordings in other brain areas and species[7,37], we found that spiking statistics of putative-pyramidal (PYR) and fast-spiking (FS) neurons exhibited large temporal irregularities, high trial-to-trial variability, heterogeneous spike rates and diverse selective responses. Distributions of firing rates of pyramidal neurons were approximately log-normal (Fig. 1B–C). Spiking activity exhibited wide inter-spike-interval (ISI) distributions, with average coefficient of variations of the ISIs that varied between behavioral states (Fig. 1B, $CV_{ISI}$ = 0.9 during sample, and $CV_{ISI}$ = 1.2 during delay; n=619, paired Student t-test $p$ = 3.39 × 10$^{-44}$), but were close to one (for comparison, neurons that fire regularly have low $CV_{ISI}$, while for a Poisson neuron $CV_{ISI}$ = 1). Importantly, the $CV_{ISI}$ was large for neurons of both low and high spike rates. The log-normal distribution of firing rates and the large $CV_{ISI}$ are in line with theoretical predictions based on the fluctuation-driven regime (e.g. see[11,13,19]). FS neurons fired, on average, at higher rates than pyramidal neurons and their rate distribution was also well-approximated by a log-normal distribution. Similarly, their

spiking activity was highly variable, with $CV_{ISI}$ that was slightly higher than the $CV_{ISI}$ of the pyramidal neurons. Neurons in both populations exhibited a high level of variability across trials[30]. This activity was selective to licking direction and was diverse across the population[35] (see also below).

We next examined membrane potential measurements from whole-cell recordings of ALM neurons (Fig. 1D–G[34,35]). To minimize the effect of movement artifacts resulting from the mouse licking during the response period, we concentrated our analysis on the delay period (mean voltage and SD during the sample and delay periods were very similar, Fig. S1A–B). Consistent with previous reports (e.g.[38]), firing rates of excitatory neurons were a bit smaller when measured using whole-cell recordings (4 ± 4$Hz$ during the delayed period) than with silicon probes (6 ± 8$Hz$), although these differences disappeared when we considered only the neurons that spiked (unpaired Student t-test, $p$ = 0.38 Fig. S1C). Similarly, average $CV_{ISI}$ of neurons in the whole-cell data were close to one, although a bit smaller for the delay period when estimated using silicon probes ($CV_{ISI}$ ≈ 0.8 during sample and during delay, Fig. S1C).

The sub-threshold fluctuations of most of the recorded neurons were well-approximated by a Gaussian distribution (excluding their spikes, example in Fig. 1D; skewness of distribution was 0.5 ± 0.44, mean ± SD; Fig. S9D[32]). The time-average voltage across the population was − 50$mV$ (range: − 64.5$mV$ to − 33.5$mV$). There was a large level of heterogeneity around this average value (Fig. 1E–G), with only a small fraction of the neurons that were close to their spike threshold (threshold was − 34 ± 3$mV$). We quantified this heterogeneity in sub-threshold statistics by the standard deviation of the time-average voltages across the population, $\Delta_V$, which was around 6$mV$ for ALM neurons. For neurons that spiked ($n$ = 35/47 neurons), when compared to their spike thresholds, the mean voltage ranged from 6$mV$ to 25$mV$ below threshold (with mean of 13$mV$ and SD of 4$mV$). Furthermore, the level of fluctuations of sub-threshold activity of neurons around their temporal-averaged value varied across the population (standard deviation of voltage fluctuations, $\sigma_V$, is in the range 1.2 − 4.5$mV$). Neurons close to threshold exhibited a larger level of voltage fluctuations. The level of fluctuations and mean voltage were positively correlated across the population (Fig. 1G).

In summary, our analysis revealed that the statistics of the spiking activity of ALM neurons in the task were consistent with previous recordings in behaving animals. Notably, the $CV_{ISI}$ of the neurons was large, and neuronal firing rate distributions exhibited wide variability, well-described by a log-normal distribution. Fast-spiking neurons demonstrated higher firing rates than putative-pyramidal neurons. Sub-threshold voltage analysis unveiled substantial diversity in time-average voltage, characterized by a large $\Delta_V$. In addition, the distributions of sub-threshold voltage fluctuations of many neurons were well estimated by a Gaussian distribution, with a positive correlation between fluctuation levels and mean voltages across the putative-pyramidal population.

### Standard spiking network models are inconsistent with the sub-threshold statistics of cortical neurons

We next asked if network models that operate in the fluctuation-driven regime could account for the spiking as well as the sub-threshold activity of neurons in the data. We considered a network consisting of excitatory and inhibitory integrate-and-fire neurons, where cells were randomly and sparsely connected to each other (Fig. 2, parameters in Table 1). The post-synaptic neuronal voltage, $V(t)$, evolved in time and changed in response to synaptic currents. These currents were modeled as $I_{syn}(t) = - gs(t)(V(t) - E)$, where $E$ is the synaptic reversal potential, $g$ was the change in synaptic conductance induced by a pre-synaptic spike, and $s(t)$ was a filtered version of the pre-synaptic spikes (see Methods for a complete description of the model). These type of network models are known in the literature as conductance-based

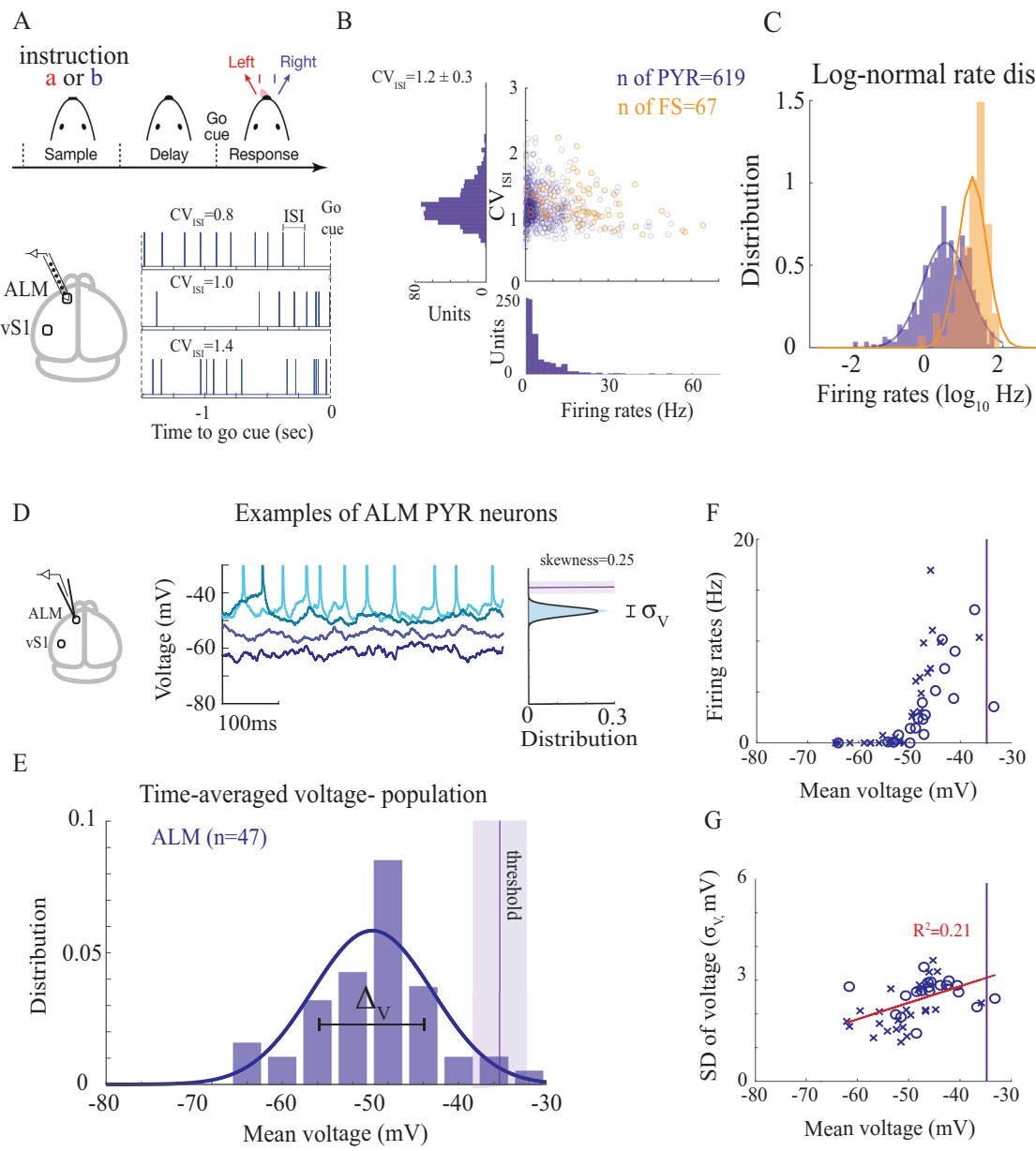

**Fig. 1 | Supra- and sub-threshold statistics of ALM neurons in mice performing a delayed-response task. A** Top: behavioral task. The instruction (tones for the auditory task, Fig. 1A–G; pole location for the tactile task, Fig. 1D–G) was presented during the sample epoch. The mouse reported its decision after the delay epoch by directional licking. The duration of the delay epoch was 1.2 s (Fig. 1D–G) or 2.0 s (Fig. 1A–C), Bottom: recording area (see also[35]) and spike trains of 3 example neurons with different coefficient of variation of the inter-spike-intervals ($CV_{ISI}$). Recordings shown in A-C were performed using silicon probes (extracellular). **B** Coefficient of variations of the inter-spike-interval ($CV_{ISI}$) against neuronal firing rates for putative pyramidal (PYR) and fast-spiking (FS) inhibitory neurons (FS). **C** Probability density function (pdf) of the log-rates is well-approximated by a Gaussian distribution (solid lines). **D** Left: cartoon of whole-cell recordings (intra-cellular). Middle: Activity of four example neurons during the first 0.5 second of delay period, together with an illustration of the inter-spike-interval. Right: Sub-threshold voltage distribution (excluding spikes) of one of the neurons. Solid line: fit to a Gaussian distribution. Purple line and shaded areas: neuronal threshold (mean ± SD). $\sigma_V$: SD of the single-neuron fluctuations. Recordings shown in D-G were performed using whole-cell. **E** pdf of mean voltage of the $n = 47$ recorded ALM PYR neurons. Solid line: fit to a Gaussian distribution. Purple line and shaded area around it: average threshold across the population and its SD ($-34 \pm 3 mV$). **F** Firing rates vs. mean voltage of ALM PYR neurons. Crosses: auditory instruction (28 neurons). Circles: tactile instruction (19 neurons) (see Methods). **G** SD of single-neuron voltage fluctuations against the mean voltage. Red line: linear regression. In all panels the data is analyzed for correct lick-right trials during delay period (see also Fig. S1). Source data are provided as a Source Data file. Figure 1A is adapted from Inagaki, H.K., Fontolan, L., Romani, S. et al. Discrete attractor dynamics underlies persistent activity in the frontal cortex. Nature 566, 212-217 (2019). https://doi.org/10.1038/s41586-019-0919-7.

networks because variations in synaptic inputs lead to changes in the input conductance of the neuron[14,39–41].

As predicted by theoretical works, the distributions of firing rates for both excitatory and inhibitory populations were well approximated by a log-normal distribution (Fig. 2B–C[19,41]). We chose the synaptic parameters such that the distributions of neuronal firing rates in the

populations fit well the spiking data of ALM neurons. In particular, the firing rate of the inhibitory population was larger than the excitatory one. With these parameters, the size of the excitatory and inhibitory post-synaptic potentials were within the physiological range (Fig. 2A). Neurons in both populations fired irregularly and with $CV_{ISI} \approx 1$, as in the data. It was slightly higher for the inhibitory population than the

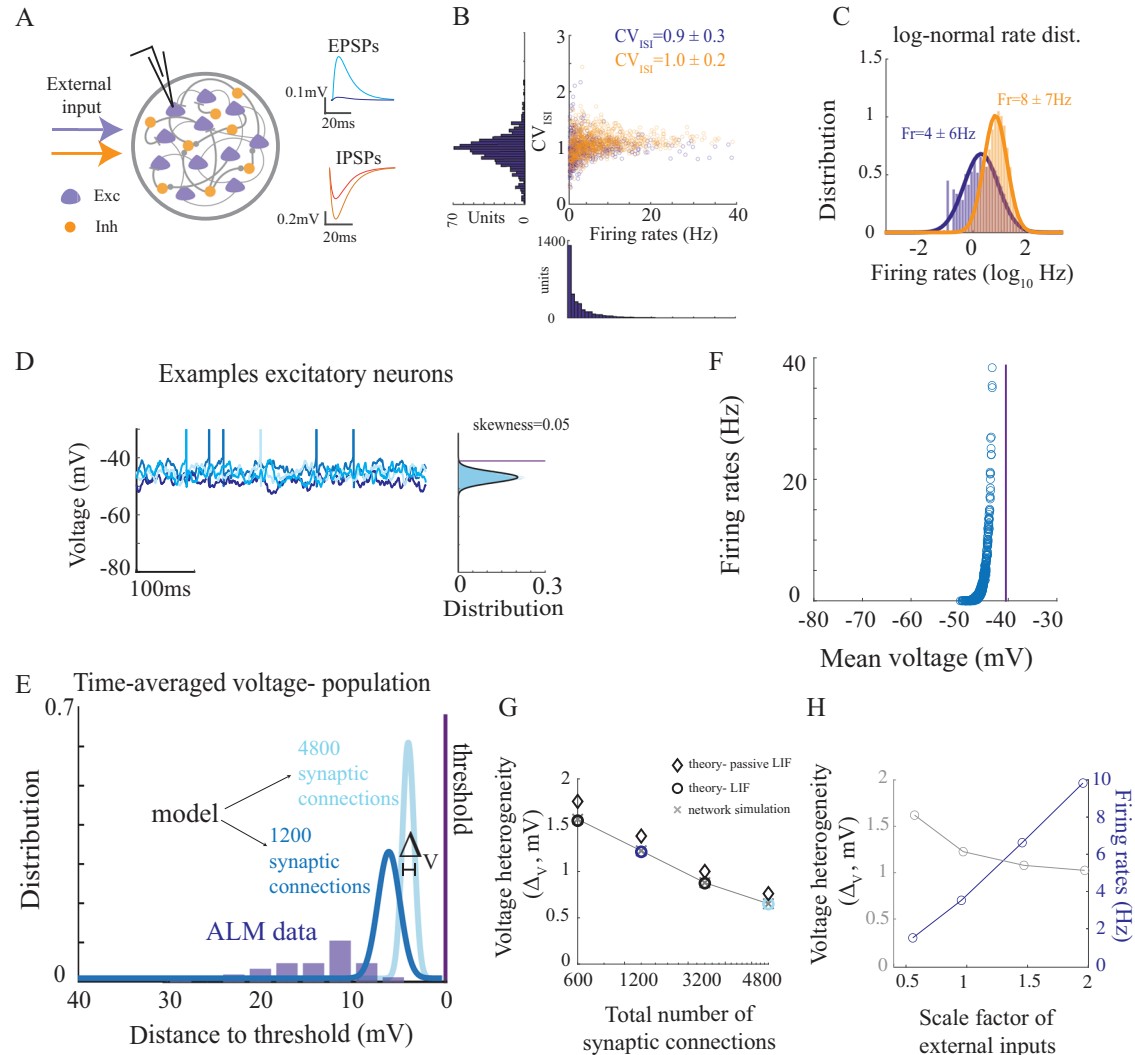

**Fig. 2 | A network of one-compartment integrate-and-fire neurons is consistent with irregular spiking statistics in ALM but fails to reproduce sub-threshold voltage statistics. A** Left: Diagram of the recurrent neural network. Synapses in the network are mediated by variations in the synaptic conductances. Right: excitatory and inhibitory post-synaptic potentials in the network. Cyan: Inh-to-Exc connection. Blue: Exc-to-Exc. Red: Exc-to-Inh. Orange: Inh-to-Inh. **B** $CV_{ISI}$ vs. firing rate for excitatory (blue) and inhibitory (orange) neurons in the network. **C** Distributions of firing rates in the network. **D** Left: Activity of four example neurons. Right: Sub-threshold voltage distribution (excluding spikes) of one of the neurons. Solid line: fit to a Gaussian distribution. Purple line: neuronal threshold. **E** Distribution of time-average voltage, plotted as the distance to threshold (purple line) of all neurons in a network with an average of 1200 (blue) and 4800 (cyan) pre-synaptic inputs per

neuron (total synaptic connections: $K_I + 2K_E$; Methods). Threshold in simulations was at $-40mV$. ALM data from Figure Fig. 1E is re-plotted here as the distance to neuronal threshold for comparison. **F** Firing rate vs. mean voltage for all neurons in the network of a total 1200 pre-synaptic inputs. **G** Voltage heterogeneity vs. number of average pre-synaptic inputs. Crosses: network simulations. Diamonds: predictions using a passive neuron (first term in Eq. (1)). Circles: predictions using an integrate-and-fire neuron, including threshold (full expression in Eq. (1)). **H** Voltage heterogeneity and firing rates of excitatory neurons when scaling the strength of external inputs by a factor. Total synaptic connections: 1200. Voltage heterogeneity stays low also for networks with rates below 2Hz. See parameters in Table 1. Source data are provided as a Source Data file.

excitatory one. This variability was self-generated by the network. The fact that the $CV_{ISI}$ was large for both neurons that fired at low and high rates is a hallmark of the fluctuation-driven regime, where fluctuations are driving the spikes[11,13].

We next investigated the sub-threshold statistics of the simulated neurons. In contrast to the spiking statistics of neurons in the network, we found that the network model failed to reproduce the sub-threshold statistics of ALM neurons. Specifically, although, like the data (Fig. S9D), the sub-threshold voltage distribution of individual neurons in the model exhibited an approximately Gaussian shape (Fig. 2D; a characteristic proposed to signify a fluctuation-driven regime[32]), the time-averaged voltage across all neurons in the model was very close to their spike threshold, exhibiting a very small standard

deviation (SD) around the average value (Fig. 2D, E; mean voltage of $-45mV$ with threshold at $-40mV$ and SD of $\Delta_V = 1mV$). Similarly, the amplitude of the temporal voltage fluctuations across the population was narrowly distributed ($\sigma_V \approx 2mV$ for all neurons). This is in contrast to the large heterogeneity in mean voltages and SDs across the ALM neurons (Fig. 2E, F, Fig. 1E, G). Thus, while accounting for the heterogeneous spiking statistics, the model failed to reproduce the heterogeneity in sub-threshold voltage activity across the population of ALM neurons.

## Narrow distribution of mean voltages in network models
How is it that firing rates in the simulated networks were so heterogeneous while the time-average voltage distribution was so narrow?

We found that this effect was a direct outcome of the regime in which the network operated. Indeed, opening of a synaptic channel increases the effective conductance– the 'leak', of the neuron. Due to a large number of open synaptic channels, the neuron becomes very leaky and its time-average voltage is clamped to an effective reversal. This effective reversal depends on the time-average conductances and reversal potentials of its excitatory and inhibitory synapses. Moreover, with voltage fluctuations of ~ 2mV, this time-average voltage must be very close to threshold in order for the neuron to fire. The fact that the firing rates are highly heterogeneous is remarkable, and it results from the large gain of the firing-voltage curve in these models[40], (Fig. 2F).

To make this argument quantitative, we studied analytically the heterogeneity in the sub-threshold activity of one-compartment integrate-and-fire neurons. We used a mean-field theory to determine the response of a population of non-interacting neurons to inputs from populations of excitatory and inhibitory neurons[40,41], (Supplementary Information). Pre-synaptic inputs were modeled as Poisson processes, with heterogeneous rate distribution (e.g., log-normal distribution). We determined the distribution of the voltage of the non-interacting neurons as a function of the number, strength and rate distribution of their pre-synaptic inputs. Using this approach, the time-average voltage of a post-synaptic neuron, $\bar{V}_i$, is (see Supplementary Information):

$$\bar{V}_i = V_{pas,i} - \tau_{pas,i}\nu_i(V_{th} - V_r) \tag{1}$$

where $\nu_i$ is the firing rate of the $i$'th neuron and $V_{th} - V_r$ is the difference between the threshold ($V_{th}$) and the reset voltage ($V_r$).

In the Supplementary Information, we analyzed Eq. (1) to show that in large models, the voltage values are very similar across all neurons (Fig. 2E, G). Specifically, the level of sub-threshold heterogeneity is a result of two terms; the first term ($V_{pas,i}$) is a result of bombarding a passive neuron with a large number of synapses that 'shunts' the soma and clamps the neurons to the same resting state. It is given by Ohm's law, that connects between the somatic conductance and currents to the membrane potential. As the total number of synaptic inputs increase, the ratio of current to conductance tends to stabilize and becomes similar across neurons (Fig. 2G, diamonds). This is because the total current and conductance also increase with the number of inputs. With a large number of inputs, the diversity of current to conductance ratios among neurons decreases in proportion to the square root of the number of synapses (see Supplementary Information).

The second term of Eq. (1) depends on the neuronal firing rate, which varies considerably across neurons in the network. Each spike results in resetting the voltage from the neuronal threshold to its reset value, contributing to the voltage a factor of $\tau_{pas,i}(V_{th} - V_r)$, with $\tau_{pas,i}$ being the neuronal effective timescale. However, this resetting effect actually decreases the level of voltage heterogeneity, as it pushes the voltage of neurons that fire at high rates back to the reset value. In other words, the negative sign before the $\tau_{pas,i}\nu_i(V_{th} - V_r)$ term implies that this term decreases the level of voltage heterogeneity (Fig. 2G, diamonds vs circles).

Taken together, the distribution of mean voltages across the population of neurons is expected to be very narrow. This is true for a wide range of neuronal firing rates (Fig. 2H), as the width of the distribution of mean voltages across the population vanishes with increasing number of synaptic inputs (Fig. 2E, G; Supplementary Information). Thus, classic network models that operate in the fluctuation-driven regime were consistent with spiking statistics in the cortex, but fundamentally failed to account for heterogeneity in sub-threshold statistics. The low level of voltage heterogeneity across neurons was an inevitable result of bombarding neurons with a large number of pre-synaptic inputs that clamped the voltage of all the neurons to more or less the same average voltage.

## Increasing sub-threshold heterogeneity in neural networks by incorporating neuronal morphology

The previous section discussed a recurrent network consisting of simplified single-compartment neurons, also known as point neurons[39–41]. This simplification led to an increase in both the total somatic currents and the somatic conductance when more synapses were activated, such that their ratio was hardly impacted by current heterogeneities (see also Supplementary Information). As a result, when many synapses were active at the same time, the membrane potential of all neurons in the network was clamped to a similar voltage. This effect is also expected to be present in networks with neurons that have active currents that mimic spike generation (e.g., as in ref. 23). However, in reality, neurons have extended spatial structures and the impact of synaptic inputs on the somatic currents and membrane conductance varies depending on their proximity to the soma[39]. We hypothesized that the narrow voltage distribution observed in the model resulted from neglecting the neuronal morphology.

Specifically, it has been established that the effectiveness of activating a synapse on the changes in somatic conductance decreases rapidly with distance from the soma, at a rate that is twice as fast as its impact on the somatic currents[42,43] (see also Fig. 3F, Fig. S2D, Fig. S3C and Methods). This means that by including the cell morphology in the models, it is likely that the total somatic conductance will not dramatically increase with the number of synaptic inputs (see also next section and Supplementary Information). Consequently, we hypothesized that when synapses are located along the dendrites, the clamping effect caused by the neuron's leakiness will be less severe compared to when all synapses are concentrated on the somatic compartment, resulting in a broader voltage distribution across neurons.

Most of the putative pyramidal cells we recorded in ALM were layer 5 neurons. Thus, to test our hypothesis, we first simulated a reconstructed layer 5 pyramidal neuron[44,45] (Fig. 3A, B), in which the spiking activities of its pre-synaptic neurons were Poisson. For each pre-synaptic excitatory (inhibitory) neuron we sampled its spike rate from the log-normal distribution of ALM pyramidal (fast-spiking) neurons (see Fig. 1B, C). We repeated this process multiple times, each time sampling different synaptic locations on the dendrites and a set of pre-synaptic neurons with different spike rates from the log-normal rate distribution. We used it to calculate the distribution of mean voltages over the population of simulated neurons. When localizing all synapses on the soma (Fig. 3A) we found that, similarly to what happens in a network of point neurons, the distribution of mean voltages across the different realizations of the pyramidal neuron was extremely narrow (Fig. 3C). Voltage fluctuations ($\sigma_V$), as well as voltage heterogeneity ($\Delta_V$) both decreased with increasing number of synaptic inputs (gray lines in Fig. 3D). However, when distributing the same number of synapses on the dendritic tree (Fig. 3B), we observed that the width of the time-averaged voltage distribution increased significantly (Fig. 3C). Both the voltage heterogeneity and the size of the fluctuations weakly depended on the number of synaptic inputs (black lines Fig. 3D). This weak dependency was not merely a result of adding distal synapses that do not affect the somatic voltage, as the distal synapses contributed both to the sub-threshold statistics. For example, with synaptic density of 0.6 synapses per microns, taking off all distance synapses (above $100\mu m$) increased the SD of the voltage of the neurons from $2.6mV$ to $3.3mV$.

These results suggest that voltage heterogeneity can be substantial in large neural networks that operate in a fluctuation-driven regime, provided that one takes into account the neuronal morphology.

## Network of point neurons that mimic an effective dendritic morphology

Encouraged by the single neuron simulations, we introduced a phenomenological network model of 'extended-like' point neurons that

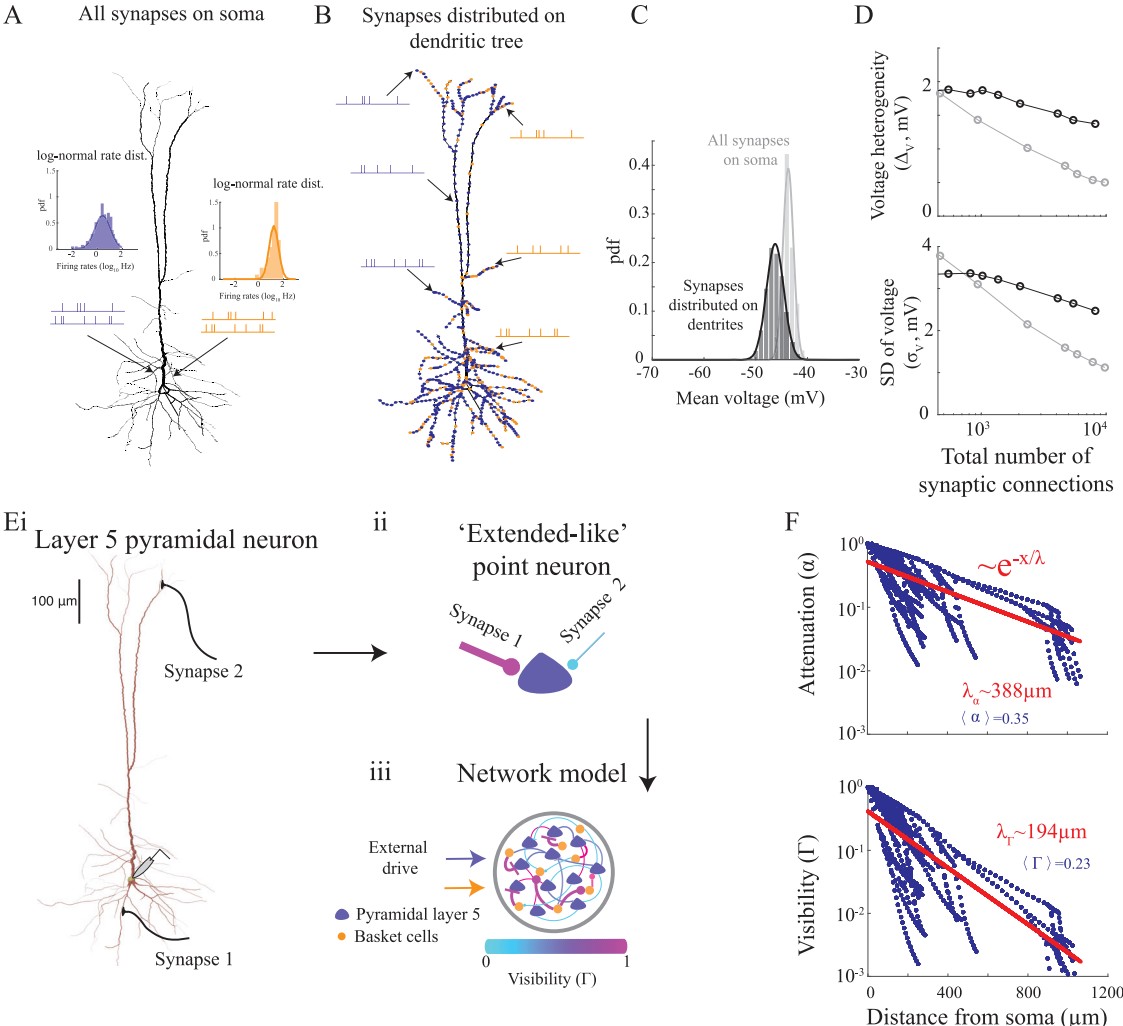

**Fig. 3 | Sub-threshold heterogeneity and the effect of distribution of synaptic inputs in a multi-compartment model of layer 5 pyramidal neuron.**
**A** Simulations of a layer 5 pyramidal neuron with all AMPA (blue) and GABAA (orange) synapses located on the soma. Pre-synaptic Poisson neurons are simulated with firing rates sampled from the log-normal rate distribution of the pyramidal (blue) and fast-spiking (orange) ALM neurons. 75% excitatory and 25% inhibitory pre-synaptic neurons are simulated for each cell. **B** Same as (A) but when synapses are uniformly distributed along the dendritic tree. **C** The probability density function of the mean voltage is estimated from simulations of multiple realizations of neurons, with 1000 realization (varying both the location of synapses and pre-synaptic firing rates) in the case of a dendritic distribution of synapses and 100 realization (varying only firing rates) in the case of somatic distribution. In both cases with ~ 2000 excitatory and inhibitory pre-synaptic neurons in total, for each neuron. Solid line: fit to a Gaussian distribution. **D** Top: Voltage heterogeneity against the number of synaptic connections. Bottom: SD of single neuron

fluctuations against the number of synaptic connections. Gray: all synapses are on the soma. Black: synapses are distributed on the dendritic tree. Synaptic conductance in C-D was 1nS. **E** Modeling a network of point neurons that incorporate the location of the synapse along the dendritic tree (see main text, Methods and Fig. S2 for details). (i) Measuring the current and conductance changes at the soma of a simulated layer 5 pyramidal neuron, when activating a synapse on a dendrite. For each synapse we extract the attenuation ($\alpha$) and visibility ($\Gamma$) parameters. (ii) An extended-like point neuron. The thicker the synapse is the larger its $\alpha$, while the visibility parameter is color coded. (iii) The empirical distribution of {$\alpha$, $\Gamma$}, estimated for each cell type, is then used to simulate a network model of spiking neurons that interact through Eq. (12). **F** Top: Attenuation vs. the distance of the synapse from soma for the cell in (E). Red line: fit to an exponential decay. Bottom: Same as top, but for the visibility parameter. Note that the decay rate of the visibility is around twice the decay rate of the attenuation parameter. Source data are provided as a Source Data file.

incorporated the effect of distributing synapses on the dendrites (Methods). Because of differences in how synaptic inputs affect the somatic currents and conductance (Fig. 3F, Fig. S2D and S3C[39,42,43]), we assumed that pre-synaptic action potentials affected the single neuron dynamics in two ways. The first was by changing the current arriving to the soma:

$$I_{soma}(t) = g\alpha s(t)(V_l - E) \qquad (2)$$

where, as before, $s(t)$ was a filtered version of the pre-synaptic spikes due to the synaptic time constant, $V_l$ was the reversal potential of the leak current, $g$ the synaptic strength and $\alpha$ was the attenuation

parameter that modeled the decay in current change with the distance to the soma. The second contribution was an effective change in the neuronal conductance:

$$g_{soma}(t) = g_l + g\Gamma s(t) \qquad (3)$$

with the leak conductance $g_l$. The visibility parameter, $\Gamma$, captured the change in somatic conductance as a result of opening of the synapse on the dendrite[43]. Thus, in this extended-like neuronal model all synapses were placed on a somatic compartment, but it effectively accounted for their location on the dendrites by changing their visibility and attenuation parameters. The case in which $\Gamma = 1$

corresponds to a pure 'conductance-based model' of a synapse, while if $\Gamma = 0$ the activation of a synapse induces a change in somatic currents but not in the neuron input conductance ('current-based model')[46,47].

In this point neuron model, as the value of $\Gamma$ decreases, the overall change in somatic conductance will be smaller, resulting in a looser clamp of the voltage membrane potential to the effective voltage. This leads to a wider range of time-average membrane potentials across different neurons. Indeed, through simulations of networks of spiking extended-like neurons, we found that using a smaller visibility parameter results in a wider distribution of mean-voltage (Fig. S5B-C). The mathematical analysis in the Supplementary Information supports this conclusion by showing that including visibility and attenuation parameters improves the ability to achieve wide voltage heterogeneity in large networks.

In order to understand how the parameters $\Gamma$ and $\alpha$ are affected by different neuronal morphologies, we simulated multi-compartment neuronal models. We estimated these parameters for each synapse by measuring the change in somatic conductance and currents when activating a synapse on a dendrite and comparing it to the same changes in the phenomenological point model. These simulations were conducted under 'in vivo conditions', assuming that the effect of a single synapse should be measured while others are active. This method is detailed in the Methods section and is shown in Fig. 3E and Fig. S2A.

Following this procedure, activation of the same synapse on the dendrite and on the 'extended-like' model led to similar activation patterns at the soma (Fig. S3, Fig. S4). As expected, the steady state response (constant synaptic activation) showed an identical maximal voltage (Fig. S3B), with only small differences in the exact synaptic response due to the voltage waveform being affected by the path of the synaptic current to the soma (Fig. S3D, Fig. S4B[48]). Furthermore, although we estimated the visibility and attenuation parameters separately for each synapse, activating multiple synapses resulted in similar voltage dynamics for a wide range of inputs, with discrepancies only appearing for high numbers of pre-synaptic inputs (Fig. S4C).

Analysis of the estimated values of the visibility and attenuation parameters showed that they decayed with the distance of the synapse from the soma, and that the visibility was always lower than the attenuation (Fig. S2D, E, Fig. 3F, Fig. S3C, see also[43]). This effect can be understood through the passive cable theory. For example, in an infinite cylinder both the visibility and the attenuation parameters decay exponentially with the distance to the soma, but the former decays twice as fast than the latter (Fig. S3C). Intuitively, this is because a current that is injected at the soma (to measure the conductance change) must propagate to the synapse and then back to the soma[43].

A detailed analysis of the distributions of $\alpha$ and $\Gamma$ highlighted their dependence on the cellular morphology. Specifically, the estimated values of $\alpha$ and $\Gamma$ for inhibitory basket cells, known for their electronic compactness compared to pyramidal neurons, exhibited larger magnitudes in comparison to the simulated pyramidal neuron's estimated parameters (Fig. 3F and Fig. S2D). The average value of the visibility parameter for the layer 5 pyramidal neuronal model fell within a range of values capable of explaining the observed large $CV_{ISI}$ and varied mean voltages in the anterior lateral motor cortex (ALM) (Fig. S5D −F; Fig. S6).

### Networks of neurons with effective dendritic morphology can account for the data and the functional consequences of the visibility parameter

Equipped with the estimated distributions of $\alpha$, $\Gamma$ for a layer 5 pyramidal neuron and for a basket cell, we next simulated a network model of point neurons that interacted through the effective synapses (Eq. (2–3) and Eq. (12)). The attenuation and visibility parameters were randomly selected from the joint distribution, $P(\alpha, \Gamma)$, which we estimated using the multi-compartmental models (Fig. 3F, Fig. S2D). Similarly to networks that neglected the neuronal morphology (Fig. 2), the spiking

statistics of the extended-like neurons resembled ALM activity (Fig. 4B–C). However, in these models, in contrast, the sub-threshold voltage distribution was wider (Fig. 4D and compare Figs. 4F and 2E) and resembled the sub-threshold voltage statistics of ALM neurons (Fig. 1E).

Specifically, neurons were no longer clamped to an effective reversal and a distribution of time-average voltage in the network was observed. Furthermore, similarly to pyramidal neurons in ALM, the mean voltage and SDs across the excitatory population in the network were positively correlated (compare blue circles in Fig. 4E with Fig. 1G). This was in contrast to the inhibitory neurons that exhibited a higher level of voltage fluctuations, which were slightly negatively correlated with their mean voltage.

This difference in correlations of mean voltage and SDs for excitatory and inhibitory neurons can be understood through a threshold effect. In particular, we derived the sub-threshold statistics for the limiting case of purely current-based models ($\Gamma = 0$) and found that these correlations originated from a non-linear effect that depended on the level of fluctuations in the inputs to the neurons (see Methods). The correlations were positive in a low current noise regime, and negative for larger noise levels (Fig. 4G). Interestingly, due to the strong recurrent inhibition in the network, the current fluctuations to an inhibitory neuron in our simulations were larger than the fluctuations to an excitatory neuron. As a result, we observed a positive correlation for the excitatory population, but a negative or small correlation for the inhibitory population (Fig. 4E; see also Fig. S7G for negative correlations in inhibitory neurons in the somatosensory cortex).

We next sought to evaluate the impact of the visibility parameter on the functionality of the ALM network. In the ALM, a substantial portion of neurons exhibits preparatory activity selective to licking direction[35]. This neuronal selectivity is diverse (across neurons) at the spike level. Similar to supra-threshold selectivity, diversity in selectivity is observed at the average voltage level as well (Fig. 5B–C). Previous studies have demonstrated that this selectivity correlates with error trials[49], with stronger selectivity in the network leading to less error trials. Therefore, we examined how the visibility parameter affects selectivity and network performance.

To investigate this, we introduced a structured recurrent motif into the initially unstructured recurrent connectivity (Fig. 5D). This introduced bistability into the network, leading to two competing populations of neurons (see[50,35] and Supplementary Information). As a result, neurons in our network model, as in the data, displayed diverse selective responses during the delay period. This selectivity was evident in their supra- and sub-threshold voltage activity (Fig. 5E, F).

In certain trials, variations in voltage dynamics, stemming from the network operating in a fluctuation-driven regime, caused switches between the two attractors (Fig. 5G–H). We interpreted these trials as error trials, reminiscent of behaviors observed in mice[35]. Notably, we discovered that the occurrence of error trials depended on the visibility parameter. Indeed, by systematically adjusting the visibility parameter in our simulations, we observed that reducing this parameter enhanced neuron selectivity. This stabilized the attractors and ultimately resulted in superior performance, characterized by a reduction in error trials (Fig. 5I).

In conclusion, our results emphasize the significance of incorporating dendrite morphology into standard fluctuation-driven models to account for both spiking and sub-threshold voltage statistics in ALM neurons. The inclusion of dendrites in the network models served a functional role by increasing neuronal selectivity, leading to a reduction in error trials and an overall improvement in task performance.

## Discussion

Spiking activity in the cortex is irregular and heterogeneous across neurons. Intrinsic recurrent network dynamics can account for these features by hypothesizing that excitatory and inhibitory inputs to a neuron are strong but they approximately balanced[2,11,15]. The net input

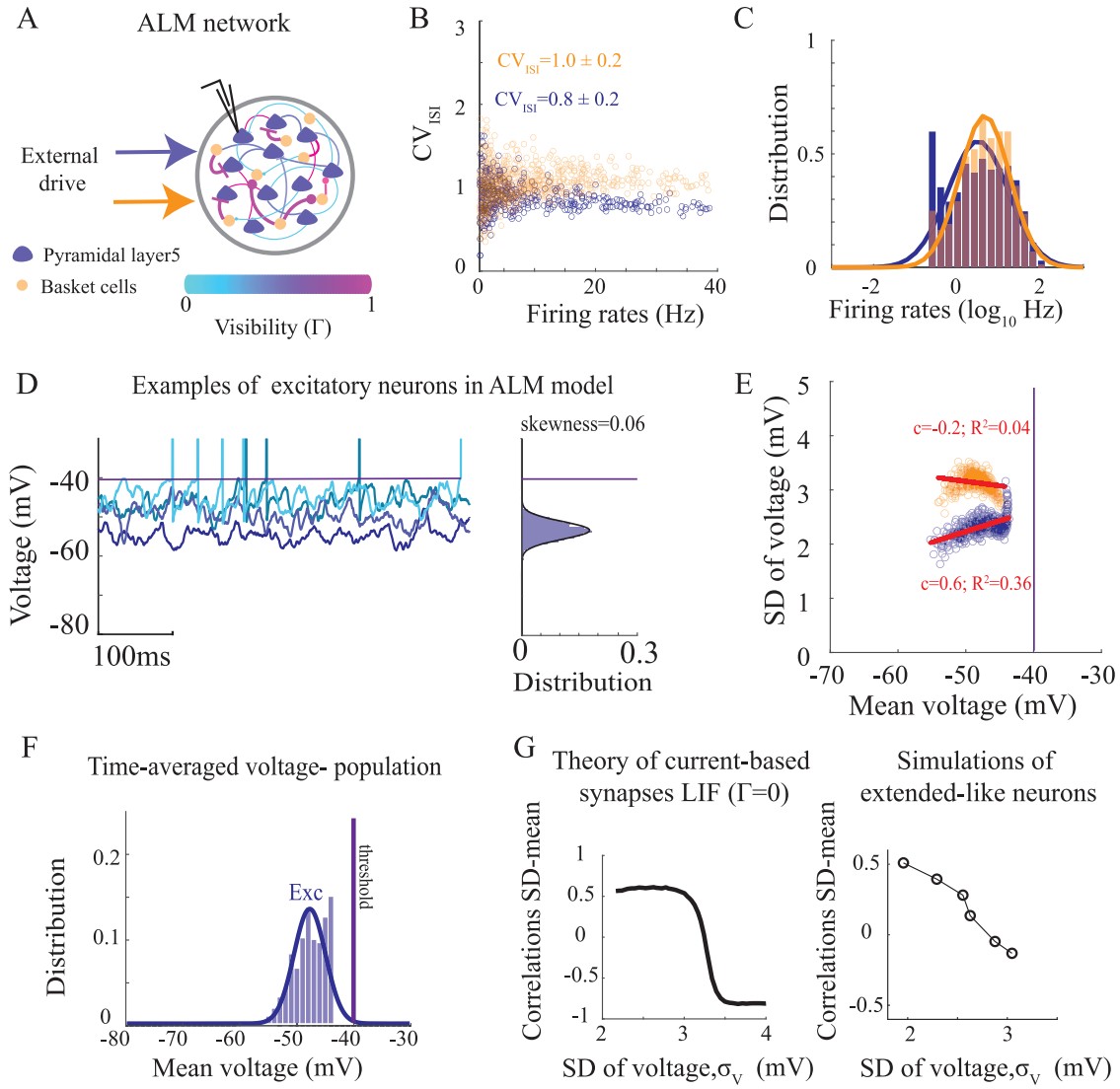

**Fig. 4 | Supra- and sub-threshold statistics of neurons in networks of extended-like point neurons. A** Diagram of an ALM network model with attenuation and visibility parameters estimated from multi-compartment models of layer 5 pyramidal (blue) and basket (orange) cells. **B** $CV_{ISI}$ vs. firing rate in the network for excitatory (blue) and inhibitory (orange) neurons. **C** Distribution of the log-rates in the network is well-approximated by a Gaussian distribution (solid lines). **D** Left: Activity of four example neurons. Right: Sub-threshold voltage distribution (excluding spikes) of one of the neurons. Solid line: fit to a Gaussian distribution. Purple line: neuronal threshold. **E** Voltage SD against mean voltage across the population for excitatory (blue) and inhibitory (orange) neurons. Red: linear fit with

a slope (c; mean-SD correlation) and goodness-of-fit ($R^2$). **F** Probability density function (pdf) of time-average voltage for the neurons in the networks. **G** Left: Theoretical prediction for the correlation between SD and mean voltage across the population in the case of current-based (Eq. (12), $\Gamma = 0$) integrate-and-fire neurons, plotted against the voltage SD (Supplementary Information, Eq. (22--23)). Right: Same as left, but for simulations of extended-like networks, as in (A-E). Each point corresponds to a network simulation with network parameters that keep the rate constant, but varies the voltage SD of the neurons. See parameters in Table 1. Source data are provided as a Source Data file.

current and its fluctuations are then of the same order. As a result, in this fluctuation-driven regime, fluctuations in synaptic inputs can drive neurons to fire action potentials irregularly with spike rates that vary across neurons[11,13,51]. Does the 'balance hypothesis' also explain the voltage statistics of the neurons?

We investigated this question by examining both the supra and sub-threshold voltage dynamics of neurons recorded from the anterior lateral motor cortex (ALM) in mice performing a decision-making task. We showed that the spiking patterns in the ALM can be explained by the presence of a large number of excitatory and inhibitory inputs, located on the neuronal somatic compartment, that balanced each other[11,19,51]. However, in terms of sub-threshold statistics, we found that the distribution of mean voltages in the data was much wider than in the model. Our simulations and mathematical analysis revealed that this was because excitatory and inhibitory inputs induced a large

increase in the neuron input conductance, leading the voltage of neurons in the model to fluctuate around the same effective reversal potential close to the threshold.

Yet, in real neurons, synapses are distributed along the dendritic tree. When we incorporated the morphology of cortical neurons into our model, we discovered that the balance hypothesis explained not only the spiking statistics but also accounted for voltage statistics in ALM. We thus propose that the ALM operates in a fluctuation-driven regime, in which excitatory and inhibitory currents are balanced.

## Sub-threshold neuronal activity in other cortical areas during behavior

Here we focused on the sub-threshold activity of ALM neurons. Distributions of average membrane potentials of excitatory neurons in ALM were wide, with an average approximately $10mV$ below their spike

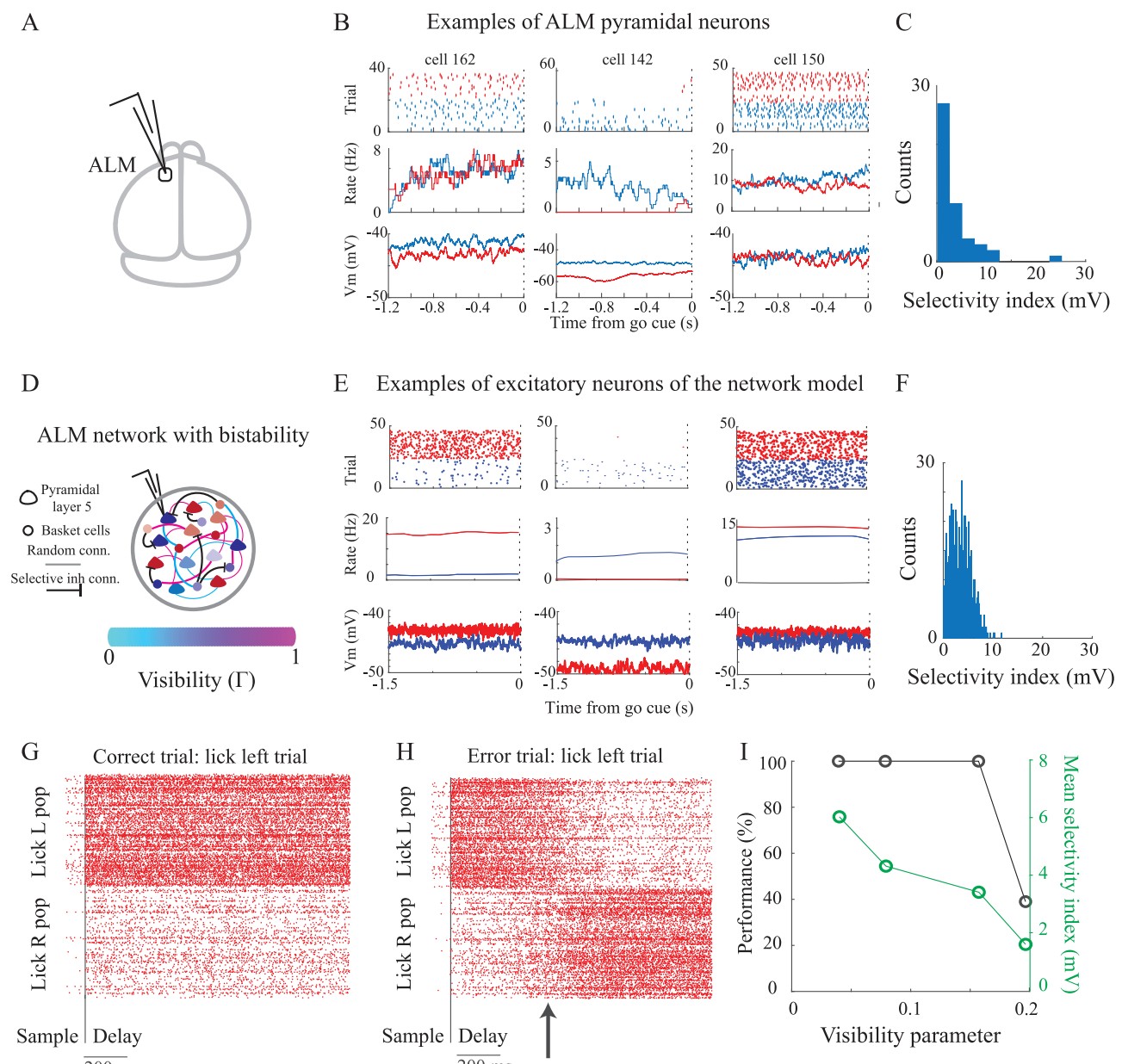

**Fig. 5 | Sub-threshold heterogeneity in selectivity of ALM neurons to licking direction and functional implications of the visibility parameter.** ALM data in (A-C) and model in (D-I). **A** Recording area. **B** Top: Raster plot of spikes for lick right (blue) and left (red) trials; Middle: Peri-stimulus time histogram (PSTH) calculated using a 10ms window; Bottom: Average sub-threshold activity (excluding spikes) **C** Distribution of sub-threshold selectivity to licking direction ($SI_i = |\bar{V}_i^R - \bar{V}_i^L|$, with $\bar{V}_i^{R/L}$ the average voltage for lick right/left of the $i$'th neuron). **D** Cartoon of a network with extended-like point neurons with visibility and attenuation parameters, estimated from multi-compartment simulations of layer 5 and basket cells. The recurrent connectivity combines a random component and a structured competitive component from the inhibitory neurons supporting bistability in the network. This bistability is the reason for the persistent and selective states in the network (see Methods and[50]). **E** Same as (B) but for examples of excitatory neurons in the model. **F** Same as (C), but for excitatory neurons in the model. **G** Raster plot of the excitatory neurons in the model. During the sample period, the input to the `lick

left' population exceeds that of the `lick right' population, signaling a `lick left' trial. Successful `lick left' trials are determined by a higher rate of activity in the `lick left' population compared to the `lick right' population at the end of the delay period. **H** Similar to (G), depicting neural activity during the delay period, but in an incorrect trial. Notice the shift in population activity from encoding `lick left' to `lick right' (indicated by the arrow). **I** Performance (percentage of correct trials) and average selectivity index of neurons in the model against the average visibility parameter. Here, we scale visibility as $x\langle\Gamma\rangle$, where $\langle\Gamma\rangle$ is derived from our multi-compartment model's average visibility estimation. By running networks with different values of $x$, we find that reducing visibility (smaller values of $x$) contributes to stabilizing attractor states, leading to stronger selectivity and better performance. The data presented in (D)-(H) corresponds to average visibility of a network of layer 5 neurons with $\langle\Gamma\rangle = 0.16$. See parameters in Table 1. Source data are provided as a Source Data file.

threshold. Notably, these findings align with similar sub-threshold patterns observed in neurons across other frontal areas during delay periods[52].

In contrast, whole-cell recordings from sensory cortical areas revealed distinct characteristics. For example, excitatory neurons in visual cortex[32] and auditory cortex[53] exhibited synchronous fluctuations, remaining approximately $20 mV$ below their spike threshold during spontaneous activity. A parallel observation in the vibrissal somatosensory area (vS1) layer 4 also indicated a significant hyperpolarization of excitatory neurons[38].

In our Supplementary Information, we extended our analysis to layer 4 (Fig. S7) and layer 5 (Fig. S8) neurons in vS1 and developed a model of the vS1 layer 4 network (Fig. S10; see Methods and Supplementary Information). Similar to ALM neurons, we found that excitatory neurons in vS1 displayed heterogeneous mean voltages, with sub-threshold fluctuations comparable to ALM neurons (average standard deviation of $2-3mV$). Notably, while vS1 excitatory neurons, especially in layer 4, were hyperpolarized[38], we found that inhibitory neurons in vS1 were considerably more depolarized than their excitatory counterparts, with a distribution of distance to threshold that was very similar to the distribution of the ALM excitatory neurons. These results are consistent with recent reports[54].

Based on these results and our model, we propose that the vS1 network operate in a 'partially-balanced' regime, were only the inhibitory population is balanced. Such a dynamical regime can emerge in network models when the ratio between the feedforward drive to the excitatory and inhibitory populations is disproportionately small compared to the recurrent inhibitory interactions[11,51] (Methods). This finding is consistent with experimental results regarding thalamic input to layer 4 neurons in the vS1 cortex[31,55,56]. Future theoretical and experimental works should explore the functional implications of this partially balanced regime, investigate its presence in other brain regions, and assess how it varies in response to inputs and varying task demands.

## Perspectives

We developed an extended point neuron model to address discrepancies with data that arose from the failure of traditional point neuron network models to explain heterogeneous sub-threshold activity. This model successfully overcame most discrepancies with the data. Through mathematical analysis (see Supplementary Information), we showed that such a model, that includes a mixture of current and conductance synapses, can produce finite conductance and current fluctuations by adjusting the average visibility of the synapses, $\Gamma$, instead of adjusting the connection strengths, as previously done in[11] or[41]. Specifically, we demonstrated that by scaling $\Gamma$ with the inverse of the number of pre-synaptic connections in the network (and thus $\alpha$ with the inverse of the square root of the number of pre-synaptic connections), we were able to ensure that both the neuronal conductance and current fluctuations were finite for large networks.

It is also worth noting that while decreasing $\Gamma$ makes the network more like a 'current-based' network, the level of heterogeneity in the network cannot be as high as desired even if the network is made up of current-based synapses. In fact, the level of voltage fluctuations, $\sigma_V$, is closely related to the heterogeneity in voltage, $\Delta_V$, through the self-consistent equations that link inputs (currents) and outputs (firing rates) of the neurons (see Supplementary Information). Thus, in recurrent networks, the level of voltage fluctuations (over time) limits the amount of voltage heterogeneity (across neurons).

Incorporating a diverse range of in-degree connectivity in network models may increase voltage heterogeneity, but it also has the potential to push the network out of a balanced regime unless additional compensatory mechanisms are introduced[57]. It was recently argued that conductance-based networks can support a wider in-degree connectivity patterns[41]. Yet, we found that incorporating large heterogeneous in-degree distributions in our conductance-based networks that fit ALM activity only slightly increased the voltage heterogeneity without deviating significantly from log-normal shape of the rate distribution (Fig. S11B–D).

Despite the advancements our model brings to more biologically-relevant network models, there are other biological factors we did not model which could further increase voltage heterogeneity, once dendrites are included. For instance, we modeled the effect of activating simple AMPA/GABA synapses in multi-compartment models on the soma, but cortical neurons consist of non-linear NMDA synapses[58,59], active non-linear ion channels that can lead to localized dendritic spike-like activity[60,61], and different mechanisms (such as voltage-gated channels and gap junctions) that may affect the input resistance[62,63]. Incorporating such active voltage-dependent conductances in network models can further increase the voltage heterogeneity beyond what is gained from the neuronal morphology and is an important future direction.

Specifically, the number of synapses required to surpass a given distance-to-spike threshold can exhibit significant variation due to non-linear NMDA currents. Once a sufficient number of synapses are activated, a substantial voltage surge can occur, even with a relatively small synapse count, owing to the regenerative properties of NMDA receptors[64,65]. This non-linear phenomenon can diversify the range of the distances from spike thresholds. Indeed, our preliminary results indicate that incorporating NMDA receptors in multi-compartment models resulted in an elevated level of voltage heterogeneity, which notably increased with the number of synapses (not shown).

Finally, recent computational studies have highlighted the important role of different GABAergic cell types in the network dynamics[28,66,67], which we did not include in our study. A future direction could be to incorporate our phenomenological point neurons, which allows to study the effect of targeting distal or proximal dendrites, in networks featuring a variety of cell types.

## Limitations

We studied whether fluctuations in a cortical network were consistent with a mechanism in which they emerged from the recurrent network dynamics. In reality, membrane fluctuations are a combination of external and recurrent fluctuations, and the relative proportion can vary between areas, populations and behavioral states. Therefore, some of the variance in voltage fluctuations could be potentially attributed to non-local recurrent connections. It would be interesting to control for such unobserved external inputs for local cortical networks while measuring the neuronal membrane potential. This will potentially require multi-brain area recordings with voltage measurements to determine the respective contributions of feed-forward and recurrent connectivity to membrane potential fluctuations of neurons.

We note that there were several factors that may have caused an overestimation of sub-threshold heterogeneity in the data. For example, previous studies have shown that the dynamics of membrane potential can vary substantially between cell types[68,69]. Although we analyzed the data separately for inhibitory and excitatory neurons, and separated the membrane potential dynamics of neurons in different cortical layers, it is still possible that our data contained a mixture of cell types. In fact, molecular studies suggest up to 100 different cell types in ALM[70]. Additionally, movements can significantly impact brain states and affect membrane potential dynamics[69,71]. We attempted to control for this by focusing on the delayed period in ALM neurons during which licking is minimal. However, we did not account for other types of animal movements (e.g. see[10]). Furthermore, we found that the mean voltage during the tactile task was slightly higher than during the auditory task, which may suggest that whisker movements affected ALM neurons (Fig. 1F). There were also several technical limitations in the recordings that could have contributed to overestimating the level of sub-threshold voltage heterogeneity (see Methods).

Furthermore, an important aspect of our research lies in establishing self-consistency between the distribution of spiking activity within the neuronal population and the distribution of sub-threshold activity. In the experimental data, the majority of supra-threshold activity is captured through extracellular recordings. It is acknowledged that this method tends to favor the selection of neurons with higher firing rates, introducing a potential bias. However, our simulations show that voltage heterogeneity remains considerably small in

models of point neurons, even when we account for lower firing rates in our modeling (Fig. 2H).

Even with the limitations acknowledged, our work suggests that in standard point neuron network models operating in the balanced regime, the variation in sub-threshold voltage is minimal and it is unlikely to find neurons deviating far from their spike threshold (Fig. 2E). This lack of diversity in voltage dynamics in standard models, when not accounting for cell morphology, contrasts with observations made in the cortex, even if the standard deviation of time-average voltage of neurons in the cortex is lower than what we reported (see also Supplementary Information).

To conclude, our work suggest that the ALM operates in a fluctuation-driven regime, in which the excitatory and inhibitory currents are balanced. This observation aligns with a recent analysis examining the responses of ALM neurons' average activity to optogenetic perturbations[30]. Theoretical works suggest that strong and balanced excitation and inhibition may be needed for neurons to generate selective responses that are robust to noise or generate stable memory states[72], as found in the frontal cortex[35,52]. Notably, the inclusion of dendrites in our extended model seems to enhance the stability of memory states (Fig. 5I). Furthermore, neurons that are driven by fluctuations in their synaptic inputs are also very sensitive to heterogeneities in the network activity and connectivity[23,24,50,73]. This feature is beneficial in networks that undergo synaptic reorganization, as it allows the neurons to develop task-related activity during learning with minimal synaptic modifications[30].

In the near future, advances in technologies will allow simultaneous recording of the sub-threshold activity of many cortical neurons (e.g.,[74,75]). Understanding the statistics of these measurements and how they vary with the dynamical state of the cortex will be a crucial step in gaining insight into the operating regime of the cortex and its functional implications.

## Methods

Simulations were done using $C^{++}$ and NEURON 7.8/8.2.[76] Analyzes of simulations and data were done using Matlab (Mathworks), Python, NumPy[77] and Matplotlib[78]. Data are presented as mean ± standard deviation (SD), unless otherwise noted.

### Behavior

Details of the delayed response task have been published previously[34,35]. Briefly, mice were trained to respond to an instruction cue by licking to one of two licking ports. For the auditory task, at the beginning of each trial, five tones were presented at one of two frequencies: 3 or 12 kHz (for lick-right and lick-left trials, respectively). The sample epoch (1.15 s total) was the time from onset of the first tone to the end of the last tone. A delay epoch followed the sample epoch and lasted for 1.2 s (for whole-cell recordings), or 2.0 s (for silicon probe recordings). An auditory 'go' cue separated the delay and the response epochs.

For the tactile delayed-response task (whole-cell recordings), at the beginning of each trial, a pole moved within reach of the whiskers. The sample epoch (1.4 s total) was the time from onset of the pole movement to completion of the pole retraction. It followed by a delay epoch of 1.2 s and then an auditory 'go' cue that separated the delay and the response epochs.

Details of the Go/NoGo behavioral paradigm have been published previously[38]. Briefly, mice were trained on a whisker-based lick/no-lick object localization task. A pole was presented in an anterior location or in one of several posterior locations. Mice licked a lick port to receive a water reward if the pole was in a posterior location and withheld licking if the pole was in an anterior position. In whole-cell recordings, either one (C2 or D2) or two whiskers (C1 and C2; D1 and D2; or C2 and C3) were spared.

### Data analysis

Spiking statistics for ALM neurons were analyzed based on extracellular recordings using silicon probes of 667 putative pyramidal neurons and 74 putative fast-spiking neurons[35]. Inter-spike-intervals (ISI) were calculated for the delay or sample periods in each of the trials and were combined together in order to calculate the coefficient of variation of the inter-spike-intervals (ISI) of the neurons, which were calculated as the standard deviation of the ISI divided by the mean of the ISI distribution.

Spiking statistics for vS1 neurons were analyzed in a similar way. They were based on juxtacellular recordings of 95 regular spiking neurons and 43 putative fast-spiking neurons in layer 4[38], and 53 regular spiking neurons and 22 putative fast-spiking neurons in layer 5[79], with some of the fast-spiking neurons that were also identified as inhibitory neurons through optogenetic tagging[79]. Inter-spike-intervals (ISI) were calculated for each non-whisking epoch and were combined together in order to calculate the coefficient of variation of the inter-spike-intervals (ISI) of the neurons.

Membrane potentials of ALM and vS1 neurons were obtained based on whole-cell recordings. The recording details for ALM neurons are given in ref. 34,35 and for vS1 neurons in ref. 38. Recordings were done in a consistent manner across studies and were not corrected for liquid junction potentials. For ALM neurons, we combined two datasets from auditory and tactile delayed response task (crosses and circles in Fig. 1F−G, respectively). We analyze the membrane potential of neurons that exhibited stationary membrane potentials during the delay period (47/89 recorded neurons). For vS1 we had 37 L4E neurons and 38 L5E neurons. The mean voltage of the five FS interneurons of layer 5 and the eight layer 4 FS interneurons in vS1 were similar and we thus combined them for the analysis.

Spike threshold was defined as the moment when $dV/dt$ first crossed 33% of its maximal value during the trial[38]. Spikes were removed by interpolating the 0.5ms pre-spike voltage with the 10ms post-spike voltage.

Mean and standard deviation (SD) of threshold were then calculated for each cell that spiked and mean and SD of the population threshold was calculated based on the cell-specific mean thresholds. To obtain the distance of ALM neurons to their spike threshold, we subtracted the mean threshold from the mean voltage of each neuron. For neurons that did not spike, we used the minimal threshold across all recorded ALM neurons as their spike threshold.

We then calculated the moments of the single-cell voltage distribution. Specifically, we concatenated all the non-whisking periods for neurons in the vS1, or during delay response periods for ALM neurons, after we subtracted the mean voltage at each epoch/period. We then calculated the SD and the skewness of these concatenated and mean-subtracted voltage traces for each neuron. To calculate the mean activity, we averaged the mean values of all epochs/periods. For neurons in vS1 we also corrected for possible drift in the mean voltage by first regressing the mean voltage per trial against the trial number, and then subtracting the slope of the regression from the voltage trace.

**Limitations in estimating voltage heterogeneity in the data.** We found that the time-average voltage of neurons had a wide distribution across neurons, with SD that could go up to 6 − 8mV. We note that this estimate was probably an upper bound for the real voltage distribution for several reasons; 1) Although we tried to control for the state of the network, for example by analyzing only non-whisking epochs or delay periods, the network state could still change over the course of recordings and between subjects. 2) There were inherent biases in the recordings. For example, in some cases the mean voltage was drifting over time. While we corrected for the drift, this might have been imperfect. 3) We used blind recordings, which were presumably

targeting the soma, however it might be that in some cases the recordings were at dendrites or axons.

## Mapping synapses on dendrites to synapses on an extended-like point neuron

**Multi-compartment single neuron model.** We used NEURON[76] (Fig. 3, Figs. S2–S4) to simulate multi-compartmental models from the morphological reconstructions of three different cells from the mouse motor cortex layer 5 pyramidal cell, visual cortex layer 4 basket cell, and somatosensory layer 4 spiny stellate cell[44,80,81] (neuromorpho IDs: NMO161366, NMO130658 and NMO02484 respectively).

Input resistances differ between in vitro to in vivo conditions, presumably due to the constant synaptic load which is much lower in a slice. For estimating $\alpha$, $\Gamma$ we simulated the model at in vivo conditions (see below), assuming that the effect of a single synapse should be measured while others are active. In contrast, for exploring the voltage distribution in Fig. 3 and Fig. S4, we simulated the model at in vitro conditions; We then activated the synapses by simulating pre-synaptic Poisson neurons with firing rates sampled from the log-normal rate distribution of the pyramidal and fast-spiking ALM neurons.

For in vitro conditions the specific membrane resistivity (Rm) for the pyramidal cell was set to 8, $000\Omega cm^2$, for the basket cell to 14, $000\Omega cm^2$, and for the spiny stellate cell to 10, $000\Omega cm^2$. This yielded an input resistance (Rin) of 108, 160, $302 M\Omega$, respectively. These values are within the experimental range measured in vitro[82–84]. For in vivo conditions, Rm for the pyramidal cell was set to $3500\Omega cm^2$, for the basket cell to $3000\Omega cm^2$, and for the spiny stellate cell to $1000\Omega cm^2$. This yielded an input resistance (Rin) of 62, 59, $45 M\Omega$, respectively. These values are within the experimental range measured in vivo[34,85,86]. When mapping the synapses to a point-like neuron (Figs. S3 and S4), we created a single compartment model with the same Rm and changed the size of the compartment to reach the same Rin as in the full model.

All multi-compartmental models had an axial resistance (Ra) of $100~\Omega cm$, and specific membrane capacitance (C) of $1\mu F/cm^2$. Unless specified otherwise, synaptic conductance in the models was set to 0.5nS, with reversal potential of 0mV and − 80mV for the excitatory and inhibitory synapses, respectively and rise and decay time constants of 0.3ms and 10ms, respectively.

## Estimating the change in somatic conductance and somatic current upon activation of one synapse in a multi-compartment model.

Opening of a synapse on a dendrite results in variations in the neuronal conductance, as measured at the soma, as well as the current flowing into the soma. The effectiveness of these changes reduces as a function of the distance of the synapse from the soma. Our goal was to construct a one-compartment model of the soma, in which all synapses are located on the same compartment, but that also effectively accounts for their location on the dendrites. To this end, we used a multi-compartment model and opened a synapse on a dendrite while measuring the conductance and current changes at the soma. We then constructed a one-compartment model and parameterized its synapses such that activating a synapse on the somatic compartment would result in the same amount of conductance and current changes as the synapse in the multi-compartment model

Specifically, let us consider a synapse of conductance $g$ located on the dendrite of a neuron. We first measured the somatic voltage change, $\Delta V_{syn}^i$, upon activating the $i$'th synapse on the dendrite (static response; Fig. S2A). We then measured the somatic voltage change, $\Delta V_l^i$, in response to an additional current step injected at the soma, $I$. We calculated the somatic conductance:

$$g_{soma}^i = \frac{I}{\Delta V_l^i}. \tag{4}$$

and defined the visibility of the synapse that captured the change in somatic conductance, normalized by the maximal possible change[43]:

$$\Gamma_i = \frac{\Delta g_{soma}^i}{g} = \frac{g_{soma}^i - g_l}{g} \tag{5}$$

with the neuronal conductance, $g_l$, measured before the activation of the synapse.

We next approximated the current arriving at the soma upon activating the synapse on the dendrite, $I_{soma}^i = g_{soma}^i \Delta V_{syn}^i$, and defined the attenuation parameter, $\alpha_i$, which approximated the decay in current, normalized by the maximal possible current that the $i$'th synapse could inject as:

$$\alpha_i = \frac{g_{soma}^i \Delta V_{syn}^i}{-g(V_l - E)} \tag{6}$$

Here, $E$ is the reversal potential of the synapse and $V_l$ the reversal potential of the leak current. Both the visibility and attenuation parameters decay with the electrotonic distance of the synapse from the soma (Fig. S3A-C).

We determined the spatial dependence of the attenuation and of the visibility parameters by performing this calculation for each synapse while changing the location of the synapse on the dendritic tree. This provided us with the empirical distribution of $\{\alpha_i, \Gamma_i\}$ for each of the multi-compartment model considered. Importantly, we found that the visibility parameter decayed with the distance of the synapse from the soma much faster than the attenuation parameter. For example, the visibility parameter in an infinite cylinder with small synaptic conductance followed $\Gamma(x) \propto e^{-2x/\lambda}$, with $\lambda$ the space constant and $x$ the distance of the synapse from the soma. On the other hand, the attenuation parameter decayed more slowly, $\alpha(x) \propto e^{-x/\lambda}$ (Fig. S3C)[43].

**An extended-like point neuron model.** For the effective one-compartment model, we considered a neuron in which its membrane potential followed:

$$C\frac{dV(t)}{dt} = -g_{soma}^i(t)(V(t) - V_l) - I_{soma}^i(t) - I \tag{7}$$

where $C$, $V_l$ are the capacitance and reversal membrane potential, $I$ was the current injected into the compartment, and where we omitted the post-synaptic index to simplify notations.

We modeled the change in neuronal conductance due to opening of the $i$'th synapse to be linear in the synaptic conductance:

$$g_{soma}^i(t) = g_l + g\Gamma_i s_i(t) \tag{8}$$

with the leak conductance, $g_l$. The change in synaptic conductance induced by a pre-synaptic spike was $g$, and $s_i(t)$ was a filtered version of the pre-synaptic spike. The visibility parameter, $\Gamma_i$, was estimated from the multi-compartment model (see above).

The effective change in current arriving at the compartment due to the opening of a synapse was modeled as:

$$I_{soma}^i(t) = g\alpha_i s_i(t)(V_l - E) \tag{9}$$

where $E$ is the synaptic reversal potential. The attenuation parameter, $\alpha_i$, which modeled the decay in somatic potentials when activating synapses that are far from the soma, was estimated from the multi-compartment model.

This choice of modeling guaranteed that opening the $i$'th synapse with its estimated $\alpha_i$, $\Gamma_i$ in the effective one-compartment model and on the dendrite of the multi-compartment model would lead to comparable changes (at a steady state) in somatic currents and

conductance in both models (Fig. S3B). Obviously, it does not account for the reduction in high-frequency response caused by dendritic filtering[48,87]. Yet, this effect is not large (Figs. S3D and S4B). In addition, the model's approximation of somatic activity when each synapse is active separately may lead to deviations from the full model when multiple synapses are active simultaneously. This is due to interactions between synapses and deviations from the estimated leak conductance. These deviations are relatively small, and occur mainly when a large number of synapses are active (Fig. S4C).

Inserting Eqs. (8) and (9) into Eq. (7), rearranging, and omitting the synaptic index to simplify notations yields:

$$C\frac{dV(t)}{dt} = -g_l(V(t) - V_l) - g\Gamma s(t)(V(t) - E)$$
$$+ g\Gamma s(t)(V_l - E) - g\alpha s(t)(V_l - E) - I \quad (10)$$

We next wrote $\Gamma = \alpha\rho$ and rewrote Eq. (10) as:

$$C\frac{dV(t)}{dt} = -g_l(V(t) - V_l) - I_{syn}(t) - I \quad (11)$$

where

$$I_{syn}(t) = g\alpha s(t)\{\rho(V(t) - E) + (1 - \rho)((V_l - E))\} \quad (12)$$

With this formulation, the parameter $\rho$ interpolated between a fully 'conductance-based' model of the synapse ($\rho = 1$), in which a pre-synaptic spike led to a transient change in the post-synaptic conductance, and a fully 'current-based' model ($\rho = 0$), in which post-synaptic conductance was independent of pre-synaptic spikes[23,46,47]. We thus refer to $\rho$ as the mixing parameter.

Finally, we note that the average mixing parameter that we estimated using the multi-compartment model only weakly depended on $g$. However, the average attenuation parameter decreased with $g$. Yet, any change in the average attenuation could be compensated through increasing the strengths of the synapses in our network models in a way that best fit the sub-threshold fluctuations. It is thus sufficient to estimate $\alpha$, $\rho$ from the multi-compartment model only for one fixed value of $g$.

### Spiking neuronal network

We consider a network of $N_E$ excitatory and $N_I$ inhibitory neurons randomly connected. We denote by $\Lambda$ the adjacency matrix of the network connectivity, defined as $\Lambda_{ij}^{ab} = 1$ with probability $p_a = K_a/N_a$ and $\Lambda_{ij}^{ab} = 0$ otherwise. Here, $a, b \in \{E, I\}$ and $i, j = 1 \ldots N_a$. Each neuron thus receives, on average, $K_E$ and $K_I$ synaptic inputs.

Neurons are modeled as leaky integrate-and-fire elements. The membrane potential of neuron $(i, a)$, $V_i^a(t)$, obeys

$$C\frac{dV_i^a(t)}{dt} = -I_{l,i}^a - I_{rec,i}^a(t) - I_{ext,i}^a \quad (13)$$

where $C$ is the capacitance, $I_{l,i}^a = g_l(V_i^a(t) - V_l)$ is the leak current whose reversal potential is $V_l$ and $g_l$ the leak conductance. Whenever the voltage reaches a threshold, $V_i^a(t) = V_{th}$, it is reset to $V_i^a(t) = V_r$.

Following the reduction procedure explained above, the recurrent input, $I_{rec,i}^a(t)$, into neuron $(i, a)$ is modeled as

$$I_{rec,i}^a(t) = \sum_{j,b} g_{ij}^{ab} \alpha_{ij}^{ab} s_j^b(t)\left(\rho_{ij}^{ab}(V_i^a(t) - E_b) + (1 - \rho_{ij}^{ab})(V_l - E_b)\right) \quad (14)$$

where $E_b$ is the synaptic reversal potential and $s_j^b(t)$:

$$\tau_{syn}\frac{ds_j^b(t)}{dt} = -s_j^b(t) + \sum_{\{t_{jb}\}} \delta(t - t_{jb}) $$

Here, $\tau_{syn}$ is the synaptic time constant and the sum is over all spikes emitted by neuron $(j, b)$ at times $t_{jb} < t$.

As in Eq. (12), the parameter $0 \le \rho_{ij}^{ab} \le 1$ interpolates the synapses from being fully conductance synapses ($\rho_{ij}^{ab} = 1$) to fully current synapses ($\rho_{ij}^{ab} = 0$). The parameter $0 \le \alpha_{ij}^{ab} \le 1$ models the attenuation of the synaptic activity as a function of its distance from the soma, with small $\alpha_{ij}^{ab}$ for synapses that are far from the soma. Note that results of multi-compartment simulations shows that $\alpha_{ij}^{ab}$ and $\rho_{ij}^{ab}$ are correlated. Yet, in the homogeneous case, for which $\rho_{ij}^{ab}, \alpha_{ij}^{ab}$ and synaptic strengths are the same for all neurons within a population (Figs. S5–6), we absorb the parameter $\alpha$ into the synaptic strengths, $g_{ab}$.

Importantly, in the mean-field limit, the sub-threshold voltage distributions only depend on the mean and variance of the changes in synaptic conductance, $g_{ij}^{ab}$, and not on their specific distributions (e.g. uniform or log-normal). Thus, for simplicity, the synaptic conductances in our model, $g_{ij}^{ab}$, depend only on the population types ($g_{ab} = \bar{g}_{ab}/\sqrt{K_b}$, with $\bar{g}_{ab}$ independent of the number of connections[11,51]). In fact, even in this case, because of the distribution of $\alpha_{ij}^{ab}$, there is a distribution of EPSPs and IPSPs values.

Finally, we model the external input, $I_{ext,i}^a(t)$, as

$$I_{ext,i}^a(t) = \left(\sqrt{K_E}\bar{I}_{ext,a} + q_{ext,a}z_i^{ext}\right)\left(\rho_{ext}(V_i^a(t) - E_E) + (1 - \rho_{ext})(V_l - E_E)\right) \quad (15)$$

with $z_i^{ext}$ being a Gaussian random variable with zero mean and unity variance and $\bar{I}_{ext,a}$ is of $\mathcal{O}(1)$[51]. Here, to simplify simulations, we modeled the feed-forward synapses with an average parameter $\rho_{ext}$ and absorbed the attenuation parameter into the input strength.

**A partially-balanced regime.** We establish the concept of the 'partially-balanced regime' as a state where the excitatory neurons remain inactive, leading to an imbalance in their inputs, whereas the inhibitory neurons maintain balanced inputs. This regime is defined mathematically by the violation of certain conditions that uphold the balanced regime. Specifically, in a partially-balanced regime, $\frac{\bar{g}_{EI}}{\bar{g}_{II}} \ge \frac{\bar{g}_{EE}}{\bar{g}_{IE}}$, akin to the balanced regime. However, in contrast, $\frac{\bar{I}_{ext,E}}{\bar{I}_{ext,I}} < \frac{\bar{g}_{EI}}{\bar{g}_{II}}$ (see also[51]).

It's worth noting that when $\frac{\bar{I}_{ext,E}}{\bar{I}_{ext,I}} = \frac{\bar{g}_{EI}}{\bar{g}_{II}}$, the excitatory neurons enter a quiescent state while maintaining balanced inputs. However, this equality requires fine-tuning of the recurrent inhibition and feedforward excitation, which in turn limits the robustness of such a dynamical state.

**Choice of network parameters.** Parameters of the spiking networks are given in Table 1. We chose the network connectivity and external drive parameters so that the firing rate distributions of the neurons in the population matched the experimental data. For the layer 4 model of vS1, we selected parameters that resulted in quiescent excitatory neurons and active inhibitory neurons. This was achieved by violating the conditions that maintain the balance regime (see above). Our simulations also confirmed that reducing the ratio $\frac{\bar{I}_{ext,E}}{\bar{I}_{ext,I}}$ led to silent excitatory neurons. This is the major difference between the parameters of the ALM model network and the vS1 layer 4 model network. Additionally, to match the estimated number of neurons in vS1 layer 4, we simulated smaller networks than those used in ALM[31,88], but found similar results even in larger networks. Interestingly, in the context of supra-stabilized networks (SSN)[89], this partially-balanced regime is linked to the phenomenon of super-saturation, in which the external input to the excitatory population is insufficient, resulting in a silent excitatory population and active inhibitory neurons[90].

To model the synchronous external drive of layer 4 vS1 network we added to $\sqrt{K}\bar{I}_{ext,a}$ in Eq. (15) another brief external drive of

**Table 1 | Network simulation parameters for figures in main text**

| | Neuron parameters | Fig. 2 | Fig. 4 (random) | Fig. 5 (bistability) |
|---|---|---|---|---|
| $dt$ | Simulation time step | 0.1 ms | 0.5 ms | 0.1 ms |
| $g_l$ | Neuronal conductance | $0.1 mS/cm^2$ | | |
| $C$ | Neuronal capacitance | $1 \mu F/cm^2$ | | |
| $V_{th}$ | Spike threshold | $-40 mV$ | | |
| $V_r$ | Voltage reset after spike | $-52 mV$ | | |
| $V_l$ | Voltage reset after spike | $-55 mV$ | | |
| $E_E$ | Reversal potential of exc synapses | $0 mV$ | | |
| $E_I$ | Reversal potential of inh synapses | $-80 mV$ | | |
| | Network parameters | | | |
| $\tau_{syn}$ | Synaptic time constant | 3 ms | | |
| $N$ | Number of neurons | 40000 | 10000 | 1000 |
| $N_E$ | Number of excitatory neurons | 32000 | 8000 | 8000 |
| $N_I$ | Number of inhibitory neurons | 8000 | 2000 | 2000 |
| $K_E$ | Average number of exc synapses to a neuron | 400 | | |
| $K_I$ | Average number of inh synapses to a neuron | 400 | | |
| $p_a$ | Connection probability | $K_a/N_a$ | | |
| $\bar{g}_{EE}$ | exc to exc synaptic weight | $0.00133 mS/cm^2$ | 0.0026 | 0.00347 |
| $\bar{g}_{IE}$ | exc to inh synaptic weight | $0.02 mS/cm^2$ | 0.039 | 0.05 |
| $\bar{g}_{II}$ | inh to inh synaptic weight | $0.133 mS/cm^2$ | 0.22 | 0.15 |
| $\bar{g}_{EI}$ | inh to exc synaptic weight | $0.08 mS/cm^2$ | 0.156 | 0.10 |
| $\alpha_{ij}^{ab}$ | Attenuation parameter | 1 | estimated | estimated |
| $\rho_{ij}^{ab}$ | Mixing parameter | 1 | estimated | estimated |
| $\bar{I}_{ext,E}$ | External input to exc | $0.00533 \mu A$ | 0.0028 | 0.0019 |
| $\bar{I}_{ext,I}$ | External input to inh | $0.00747 \mu A$ | 0.006 | 0.004 |
| $q_{ext,E}$ | Disorder in external input to exc | $0.008 \mu A$ | | |
| $q_{ext,I}$ | Disorder in external input to inh | $0.005 \mu A$ | | |
| $\rho_{ext}$ | Average mixing parameter of ext inputs | 1 | 0 | 0 |
| $g_s$ | Selective inh-to-inh ratio | 0 | 0 | 5 |
| $\bar{I}_{ext,E}^1$ | External input during the delay period to exc | 0 | 0 | $0.00045 mS/cm^2$ |
| $\bar{I}_{ext,I}^1$ | External input during the delay period to inh | 0 | 0 | $0.0006 mS/cm^2$ |

$\sqrt{K}\bar{I}_{ext,a}^1$, with $\bar{I}_{ext,E}^1 = 0.012 \mu A$ and $\bar{I}_{ext,I}^1 = 0.012 \mu A$, for 10ms, which instantaneously drove the excitatory neurons also to be in a balanced regime.

**Network architecture for a balanced network that supports persistent activity through bistability.** Neurons in ALM tend to exhibit persistent, or slow, dynamics during the delay period[35]. To model this, we added to the random connectivity of ALM network model additional structured connections as in[50]. These additional structured connections allowed the balanced network to be bistable without breaking the EI balance.

We follow[50] and model an additional selective feedforward input into each population, excitatory or inhibitory, that consists of two subsets of neurons, namely left (L) and right (R) selective neurons. Before the instruction appears this additional feedforward input, $I_{S,i}^a(t)$, is zero for all neurons. Upon presentation of a right stimulus, $I_{S,i}^a(t)$ into R-selective neurons is stronger than for the L-selective neurons, and vice versa, while during delay period this input is constant and equal to both L and R populations. Specifically, instead of Eq. (15) we take

$$I_{ext,i}^a(t) = \left(\sqrt{K_E}\bar{I}_{ext,a} + q_{ext,a}z_i^{ext} + I_{S,i}^a(t)\right)\left(\rho_{ext}(V_i^a(t) - E_E) + (1 - \rho_{ext})(V_l - E_E)\right)$$

with the input

$$I_{S,i}^a(t) = \bar{I}_{ext,a}^1(1 + \epsilon\Theta(t - T_1)\Theta(T_2 - t))$$

that is non-zero only during sample and delay periods (that last $2100 ms$). The parameter $\epsilon$ is positive for R trials and negative for L trials, $\Theta(x)$ is the Heaviside function, $T_2 - T_1$ is the sample duration ($100 ms$ in simulations) and with $\bar{I}_{ext,a}^1 = \mathcal{O}(1)$. Thus, there is no selective feedforward input to neurons in the model during delay period.

We denote the subset of R-selective (L-selective) neurons in population $a$ by $R^a$ ($L^a$). Neuron $(i, a) \in R^a$ if $i = 1,...,\frac{N_a}{2}$ and $(i, a) \in L^a$ if $i = \frac{N_a}{2} + 1,...,N_a$.

The recurrent connectivity in this extended model has two components. One is functionally specific and the other is not. The non-specific component is fully random, Erdős-Rényi (ER) graph, and does not depend on the selectivity of the pre- and post-synaptic neurons. The competition between the left (L) and the right (R) selective neurons is mediated by an additional set of connections. These connections are specific and are much less numerous, but stronger than the unspecific ones. There are no specific connections from the excitatory neurons to other neurons. The probability for a specific connection from an inhibitory neuron to another excitatory or inhibitory neurons is $p_\alpha^S = \frac{2\sqrt{K_I}}{N_I}$. Therefore, each neuron (excitatory as well as inhibitory) receives, on average, $\sqrt{K}$ connections from inhibitory neurons whose selectivities are different from its own, and on average $K$ non-selective inhibitory connections.

The strength of the specific connections depends solely on the neurons' type, $g_{al}g_S$, with $g_S$ determining the strength of the selective synapses with respect to the non-selective, random, connections. It is non-zero only in the simulations of Fig. 5. The total current into neuron $(i, \alpha)$ due to the recurrent interactions follows Eq. (14), with the recurrent connections given by $g_{ij}^{al} = (g_{al}\Lambda_{ij}^{al} + g_Sg_{al}\tilde{\Lambda}_{ij}^{al})$, with $\Lambda_{ij}^{al}$ the adjacency matrix of an ER graph with $p_a = \frac{K_a}{N_a}$, and $\tilde{\Lambda}_{ij}^{ab}$ the adjacency matrix of another ER graph with $p_a^S = \frac{2\sqrt{K_a}}{N_a}$.

With this architecture and feedforward input both the excitatory and inhibitory neurons in the network are selective to the licking directions, both during the sample and without selective feedforward input during the delay period.

## Reporting summary

Further information on research design is available in the Nature Portfolio Reporting Summary linked to this article.

## Data availability

Source data are provided with this paper. Electrophysiological data for vS1 neurons (see[79]) in NWB2.0 format (https://neurodatawithoutborders.github.io/) are available at Figshare (https://figshare.com; https://doi.org/10.25378/janelia.8869115) and for ALM neurons (see[35]) at (https://doi.org/10.25378/janelia.7489253). Source data are provided with this paper.

## Code availability

Code is available at https://github.com/orena1/sub-threshold-statistics(https://doi.org/10.5281/zenodo.12534519).

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

## Acknowledgements

We would like to thank Larry Abbott, Sandro Romani, Tim Vogels, Alessandro Sanzeni and Nelson Spruston for their valuable feedback. This work was supported by the Howard Hughes Medical Institute.

## Author contributions

R.D. and O.A. conceived the research, ran simulations and analyzed the data. R.D. did the analytical calculations. H.I., J.Y. and K.S. designed the experiments. H.I. and J.Y. collected the experimental data. O.A., R.D., and K.S. wrote the paper, with inputs from H. I. and J.Y.

## Competing interests

The authors declare no competing interests.
