## [Peer Review File · Nature Communications]

Sub-threshold neuronal activity and the dynamical regime of cerebral cortexREVIEWER COMMENTS

Reviewer #1 (Remarks to the Author):

Amsalem et al. investigated if recurrent network models operating in a fluctuation-driven regime can explain cortical dynamics. They found that such models fits well with spiking data from ALM in frontal cortex but not with data from vS1. Furthermore they found that including dendritic morphology into the standard models enabled models to account for not only the spiking activity, but also the diversity in sub-threshold activity found in frontal cortex.

The inclusion of dendritic conductances is interesting and well argued in the paper. It is less clear what the analysis of activity from vS1 and comparison with data from ALM brings of conceptual advance.

- The abstract states 'However, it is still unclear whether the cortex operates in such a regime'. The question of whether fluctuation-driven dynamics is the driver of irregular spiking patterns etc. is very interesting, but it seems already clear that cortical circuits can have dynamics not in a fluctuation-driven regime - for instance in vS1. It is not clear to me what the analysis of vS1 data in its current state adds to this paper.
- Diving further into the differences between the balanced state of inhibitory and excitatory neurons could be interesting but would likely require a dedicated study based on more than 13 neurons.
- The experimental literature related to the dynamics of vS1 should be discussed.
- A direct comparison between data from ALM and vS1 is difficult but it would be very helpful to the reader to add a bit more detail about the experiments for example in the beginning of the results section and fig 1A (for instance a description of the 'instruction cue' and recording technique). A description of the instruction cue is also missing from the methods.
- Please comment on the influence of the used recording techniques. Silicon probes will have a bias towards neurons with at least some activity whereas juxtacellular recordings will be less biased. Could that have an influence on the findings?
- Please comment on the influence of the task design. In the vS1 dataset the whiskers are used for identifying the position of a pole. Could a more rich stimuli where the whiskers are exploring a complex object lead to another dynamical state of vS1?
- 'closer to the periphery' (L68) could include cortical neurons only a few synapses away from motor neurons which, I suppose, is not what is meant in this context.

The authors should consider to leave out the analysis of vS1. In my opinion the paper would be stronger with a more clear focus on how standard models fail to account for certain aspects (diversity in sub-threshold voltages) of data otherwise found to be in a fluctuation-driven regime and on how adding dendritic conductances can remedy this.

Reviewer #2 (Remarks to the Author):

Neurons emit action potentials with irregular patterns, which has significant implications for the coding schemes neural circuits use to process information. Understanding the source of this variability has been a key target of neuroscience research for decades. One explanation for this variability, which is particularly relevant to cortical networks, is based on theoretical studies that show how recurrent networks of excitatory and inhibitory neurons can reach a balanced regime. In this regime, the excitatory and inhibitory drives match, and action potentials are driven by fluctuations around a mean value. This mechanism elegantly explains how the observed variability can arise in deterministic networks.

In the current manuscript, Amsalem et al. analyzed electrophysiological recordings obtained from the sensory and motor cortices of mice during behavior. The researchers first characterized the spike trains of these neurons and demonstrated their high variability over a wide range of firing rates. They then attempted to match the balanced-network model to the data and found a mismatch. The effective conductance of each neuron increased with the number of incoming synaptic inputs, which affected its integrative properties. Consequently, the neuron's membrane potential was clamped to a value determined by the balance between the excitatory and inhibitory drives, and the amplitude of the fluctuation could not be large enough to drive the spikes while providing the required variability. The authors hypothesized that placing synapses on the dendrites rather than on the soma could partially decouple the contribution of the synapse to the membrane potential and the effective conductance. This would enable a better fit between the model and the data. The researchers demonstrated this phenomenon for a single neuron using a full compartmental model and subsequently with a network model having phenomenological synapses that enabled the decoupling. Lastly, they explained the difference in behavior between sensory neurons in layer 4 and other neurons by differences in the external synaptic drive (i.e., the non-recurrent one).

This study is a beautiful work that uses rare high-quality data to study a long-standing theoretical question. The simulations and analytical work are done to the highest-level paying attention to small but important details.

Major comments:

1. Re: “inevitable” in line 222 - A major factor in the discussion around figure 3 is the vanishing second term of equation 1 due to the reduction in the membrane time constant resulting from bombardment of the neuron by synaptic inputs (see also Rapp Segev and Koch 96). This has a very clear testable prediction of the apparent time constant of the recorded cells presented in figure 1 and 2. The authors are likely to have experimentally recorded the neurons’ responses to brief current pulses or long current pulses (standard procedure). They, thus could fit them with multiexponential functions to estimate the effective time constant of the membrane (as done previously by the first author). It is critical to see if indeed the distribution of the experimentally derived time constant is as predicted by the model assumptions.

2. It has been suggested that increase in average conductance by synaptic inputs is compensated by decrease in other currents such as I_h (see Waters and Helmchen 2006) or by deactivating potassium current I_A (Hoffman et al. 1997). Such mechanisms can be implemented in the integrate and fire neurons and might provide a solution that does not require the suggested dendritic mechanisms. It is likely that there is some variance in the expression of these current (I_h is particularly easy to spot). Is there any correlation between the CV of the neuron and the expression of I_h/I_A ?

3. There seem to be one caveat in the current story. Moving the synapses away from the soma indeed should decouple their effect on conductance from the contribution of current. Moreover, while the analysis of the visibility of the synapse is done for each synapse individually, in fact it should be done when the conductance of the membrane of the tree is set to include the average conductance of the background activity. The consequences of these two factors should be that fluctuations at the soma are mostly driven by proximal synapses. Therefore, isn’t the fact that the variability is only slightly dependent on the synapses number in the dendritic case (e.g., Figure 4D) reflects that the fluctuations are driven by the relatively fixed number of proximal synapses rather than what you suggest?

4. NMDA synapses: Another source that can contribute to the decoupling of the synapses contribution to current and conductance is non-linear currents such as the contribution of NMDA receptors. These currents are slow and can provide a lot of charge with smaller contribution to the conductance (see for example Cook and Magee 1997 & 1999 and many others), even without considering dendritic NMDA spikes. Therefore, it is possible that the resting membrane potential is boosted by non-linear currents, requiring much less conductance and hence less restrictions on the fluctuations. While NMDA currents are mentioned in the discussion, the mechanism by which they can affect the results is only poorly discussed. Please improve.

Minor comments:

1. Lines 74-78 and figure 1B it is not clear if all of these recordings are from intracellular recordings or a mixture of extra and intracellular recordings. Please specify. If there is a mixture it would be good to present the distributions separately (at least in a supp figure) because the level of confidence in spike sorting is different. Figure A suggests it is all intracellular but the numbers seem quite high for that (so the figure is a bit misleading).
2. Figure 1B it would be good to show representing example traces of cells with different CV (say 0.5, 1, 2).
3. Figure 1C – What do you make of the log-normal distribution of firing rate? Is it just an observation or does it indicate something? Is it important for your future modeling?
4. Line 98 – Are the statistics of the ISI distributions consistent with those shown in Figure 1B? This is critical!
5. Line 121 – This sentence does not make sense.
6. Line 127-128 Fig 2E does not show difference between L4 and L5. And the whole discussion in this paragraph is very confusing because you compare at least four groups of neurons (L4E, L4FS L5E (of vS1) and then also ALM (which layer?), sometimes doing different comparisons in the same sentence. Please re-write this paragraph.
7. The use of V_* in Equation 1 notation is a strange choice. Why define a variable with a strange notation that you have to explain when you can use a standard notation of (V_{th_Vr}) which makes much more sense.
8. The title: “Increasing sub-threshold heterogeneity in the network by including the morphology of the neuron” could be improved (e.g. neuron’s morphology)
9. The match between figure 1C and figure 5C is pretty poor, can you comment on that?

Reviewer #3 (Remarks to the Author):

Summary:

The manuscript of Amsalem et al. investigates spiking and voltage sub-threshold neuronal dynamics, in the sensory and frontal cortex, in mice performing decision-making tasks and their underlying network mechanisms in a novel theoretical framework of 'extended-like' point neural networks. Few experimental studies have characterized sub-threshold voltage statistics in behaving animals. Understanding these statistics is fundamental to acquiring mechanistic insight into the neural mechanisms underlying cortical activity. It is also critical for advancing hypotheses on the operating regime of cortical networks.

Firstly and consistent with previous experimental studies, the authors report that neurons in their data sets exhibit Poisson-like, temporally irregular spiking activity, with rate distributions well-approximated by log-normal distributions. Secondly, the authors show that sub-threshold neural activity in the anterior lateral motor cortex (ALM) and the vibrissal somatosensory area (vS1) share common features. In particular, neurons exhibit high voltage variability and considerable mean voltage heterogeneity, with some neurons, on average, much closer to their threshold than others.

Poisson-like cortical spiking statistics is a hallmark of strongly recurrent neural networks operating in the 'fluctuation-driven' (or 'balanced regime') in which fluctuations in membrane potentials emerge from the recurrent dynamics. Nevertheless, the authors show that the standard fluctuation-driven model fails to capture the heterogeneity in sub-threshold activity in the data. They, therefore, developed a novel phenomenological network model of extended-like point neurons with synapses that mimic dendritic integration. Their work challenges classical network models by introducing an essential element of neuronal biophysics that is too often overlooked. It is thus a significant step forward in understanding how intrinsic neuronal morphology and synaptic integration can shape population dynamics in sensory and memory systems.

The authors show how extended-like point neural networks operating in the fluctuation-driven regime allow for heterogeneous sub-threshold neural activity to emerge. Moreover, they provide a rigorous mean-field theoretical description of how extended-like features - mainly, attenuation that captures the decay in current change with the distance to the soma and visibility that captures the change in somatic conductance as a result of synapse opening - shape the network dynamics and the individual neuronal properties.

Altogether, their theoretical work points to ALM operating in a fluctuation-driven regime. In contrast, the model suggests that excitatory neurons in layer 4 of the barrel cortex are not fluctuation-driven but are driven by occasional synchronous inputs.

General comments:

The manuscript is essential to the literature on cortical neuronal dynamics and their operating regime. It is a fine example of how a theoretical approach can gain from including more biological realism in its carving and reciprocally. By varying the parameters of their extended synapses, the model can infer the voltage and spiking statistics of different cell types across different cortical layers or areas based on morphology. Even if the proposed ‘extended-like’ framework is not biological per se but is an elegant and powerful means to uncover how neuronal morphologies can impact network dynamics.

The following points are left to the appreciation of the authors.

The manuscript emphasizes the theoretical aspects of the model (i.e., extended-like modeling framework) before its functional applications (i.e., emergence of sub-threshold selectivity to licking direction in the bistable network). An essential novelty of this model is the ability to exhibit double well attractor dynamics with extended-like neurons. It is not the primary motivation of the study that is already substantial and fine as is. However, the authors may explore how sub-threshold voltage heterogeneity (e.g., by varying the visibility parameter) might affect network performance in a possible revision. If sub-threshold heterogeneity links to behavior in the model, it could be tested in the incorrect trials in the data, thus giving the model predictive power.

Finally, and on the same line of thought, reference [34] contains data with optogenetic perturbations of ALM. Optogenetic perturbations could switch ALM from operating in a fluctuation-driven regime to a partially balanced state. The author could choose to analyze and show the level of sub-threshold heterogeneity in perturbed trials and reassert the potency of the model by showing similar effects in their simulations.

A. Mahrach

Sub-threshold neuronal activity and the dynamical regime of cerebral cortex

Oren Amsalem, Hidehiko Inagaki, Jianing Yu, Karel Svoboda, and Ran Darshan

Answer to the Reviewer's comments

We thank the reviewers for their insightful and constructive reviews. We have revised our manuscript in response to their comments and addressed all the reviewer's concerns.

We would like to highlight several newly added results and discussion points that address the major comments raised by the reviewers. We respond to the reviewers' main concerns by:

- (1) Providing additional simulations that further support the conclusions of the paper, even in cases where spiking statistics may have been biased due to extra-cellular methods.
- (2) Explaining the significance of comparing vS1 and ALM data in our study.
- (3) Commenting on the limited dataset of whole-cell recordings for the fast-spiking neurons.
- (4) Including simulations and providing commentary on the effects of incorporating voltage-gated channels, such as Ih and NMDA currents, on our conclusions.
- (5) Clarifying that we measured visibility and attenuation in the multi-compartment model under 'in-vivo' conditions and demonstrating the relevance of distal synapses to subthreshold activity.
- (6) Providing additional simulations and figures that shed light on the functional implications of the visibility parameter on the selectivity levels of ALM neurons, on the stability of the attractor dynamics, and consequently, on the network performance.

In addition, we provided detailed answers to various comments raised by the reviewers, either by performing additional data analysis, adding more simulations, providing new figures, or by rewriting the manuscript accordingly. We believe that the revised manuscript provides complete and rigorous answers to all the major and minor comments by the reviewers. We respond in detail to individual points below. Reviewers' comments are set in **BLACK** and our replies in **BLUE**.

In the revised manuscript, edited and newly added texts are highlighted in **BLUE**.

Reviewer #1 (Remarks to the Author):

Amsalem et al. investigated if recurrent network models operating in a fluctuation-driven regime can explain cortical dynamics. They found that such models fits well with spiking data from ALM in frontal cortex but not with data from vS1. Furthermore they found that including dendritic morphology into the standard models enabled models to account for not only the spiking activity, but also the diversity in sub-threshold activity found in frontal cortex.

The inclusion of dendritic conductances is interesting and well argued in the paper. It is less clear what the analysis of activity from vS1 and comparison with data from ALM brings of conceptual advance.

- The abstract states 'However, it is still unclear whether the cortex operates in such a regime'. The question of whether fluctuation-driven dynamics is the driver of irregular spiking patterns etc. is very interesting, but it seems already clear that cortical circuits can have dynamics not in a fluctuation-driven regime - for instance in vS1. It is not clear to me what the analysis of vS1 data in its current state adds to this paper.

We appreciate the reviewer for finding the inclusion of dendritic conductance interesting and well-argued. While we agree with the reviewer that our work suggests that vS1 excitatory (but not inhibitory) neurons are not fluctuation-driven, it is not clear to us which papers in the literature they refer to. The relevant papers we are aware of, in which vS1 data is analyzed in the context of a dynamical regime (fluctuation-driven or not) are Hires et al. 2015 and Gutnisky, et al 2017. Specifically, the first paper argues that irregular spiking of spiny stellate cells can be explained by variability of the behavior and is not driven by the network, while the second paper argues that spiking of vS1 layer 4 (both excitatory and fast-spiking) neurons can be explained using the fluctuation-driven regime. In contrast, we argue that the large distance to threshold of the spiny stellate cells, together with their voltage SD and the spiking and voltage statistics of the inhibitory neurons, suggests that layer 4 of vS1 is only partially-balanced, consistent with Hires et al.

In any case, in our abstract we say that it is unclear if the cortex operates in 'such a regime', where we refer to the fluctuation-driven regime. Even if there is evidence that cortical circuits can have dynamics not in a fluctuation-driven regime (and there is some–debated– evidence in auditory and visual cortices), it doesn't answer the question if cortex operates in a fluctuation-driven regime.

We believe that the analysis of vS1 data, and especially its comparison with ALM, significantly contribute to the ongoing debate concerning the operating regime of the cortex (see e.g. Ahmadian & Miller 2021), encompassing several critical aspects:

Comparative analysis: One goal of our study is to compare cortical regions, layers, and cell types. Our analysis and computational modeling involve a comparison of: 1) vS1 and ALM excitatory neurons, 2) excitatory and inhibitory neurons within vS1, and 3) excitatory neurons between layers of vS1. All these comparisons were conducted using data from animals engaged in decision-making tasks..

Subthreshold and suprathreshold dynamics: The crux of our analysis revolves around combining subthreshold and suprathreshold statistics, while also integrating a computational model to elucidate the observed data in each specific cortical area. To the best of our knowledge, this is the first study that considers the subthreshold statistics of neurons (together with their suprathreshold statistics) in a self-consistent way, under the hypothesis of a fluctuation-driven regime. This approach enables us to offer new insights into the different operating regimes within the cortex. The fact that the fluctuation-driven regime hypothesis cannot explain the subthreshold voltage statistics in vS1 in a self-consistent manner, even when dendrites are included, is important.

Insights: Our findings suggest that the vS1 layer 4 circuit operates within a partially-balanced regime, a regime where the excitatory neurons are *unbalanced*, while the inhibitory neurons are

balanced. This is a novel concept that we introduce based on our data. This proposed regime represents a significant departure from previously characterized dynamical regimes. Our work provides quantitative support and computational modeling to back up this hypothesis. The comparison between the two brain regions, and between the excitatory and inhibitory neurons within vS1, thus reveals that within a similar task (almost 'at the same time'), two brain regions can operate in different dynamical regimes, and that there is a new dynamical regime of a partially-balanced state that the community should consider. This is very different and goes beyond the reviewer's statement that 'it seems already clear that cortical circuits can have dynamics not in a fluctuation-driven regime', as well as from the ongoing debate of how tight the balance in cortex is (Ahmadian & Miller 2021).

Balanced Vs. Unbalanced: The focus of our analysis of the distribution of the distance to threshold and the difference between the vS1 regime and ALM clarifies the distinction between what we expect to see in a balanced network at the subthreshold level. For example, an area can be driven by fluctuations even if its voltage is far from the threshold, if the standard deviation (SD) of the voltage is large enough. Alternatively, if the voltage SD is small and the distance to threshold is small as well, the neurons can still be balanced and fire at very low rates. The comparison between excitatory neurons of vS1 and ALM shows that their SD is similar, but their distance to threshold is very different. vS1 layer 4 neurons thus fire as a result of a strong change in their mean inputs, not due to irregular spiking patterns. While some of these statements were argued before based on the spiking statistics (Hires et al. 2015, Gutnisky, et al 2017; Yu et al 2019 etc'), the analysis of the subthreshold activity in the context of the computational model sheds new light on this statement. Also, the comparison with ALM in a similar task challenges prevailing notions that the cortex is either 'balanced' or 'unbalanced.'

We rewrote the paragraph on vS1 data to clarify the comparison with ALM data. We also emphasized that vS1 data, and the differences with our models, led us to suggest that it operates in a novel dynamical regime— a partial balanced regime. For example, we added the following paragraphs to the main text:

“...Thus, while accounting for the spiking statistics, the model failed to reproduce the distribution of mean voltages in the ALM data.”

“In contrast to ALM neurons, we found that the excitatory neurons in layer 4 of vS1 were quiescent and much more hyperpolarized, with neurons that were around 20mV below their spike threshold. This contrasts with neurons that operate in a fluctuation-driven regime that typically hover close to their spike threshold (Fig.5D). “

“Specifically, we posit that while ALM operates in a fluctuation-driven regime, characterized by a balance in both excitatory and inhibitory sub-networks, layer 4 of the barrel cortex exhibits a state of partial balance (see Methods). Within this regime, the external excitatory currents and the recurrent inhibition within the inhibitory population counterbalance each other, whereas the same balance isn't achieved for the excitatory population. Consequently, excitatory neurons can operate significantly below their threshold, while the inhibitory neurons remain close to their threshold.”

In addition, we rewrote the following paragraphs of the discussion accordingly:

“...We investigated this question by examining the neural dynamics of the anterior lateral motor cortex (ALM), as well as on the vibrissal somatosensory area (vS1) in mice performing decision-making tasks. In terms of sub-threshold statistics, we identified two types of disparities between standard spiking models and the data. First, the distribution of mean voltages in the data were much wider than in the model. Second, the distance of vS1 excitatory neurons from their thresholds was considerably larger than in networks operating in a fluctuation-driven regime, where neurons tend to remain in close proximity to their spike threshold.

However, when we incorporated the morphology of cortical neurons into our model, we discovered that the balance hypothesis explained not only the spiking statistics but also accounted for voltage statistics in ALM. In contrast, our work suggested that in layer 4 of vS1, the inputs to excitatory neurons were not balanced, while fast-spiking inhibitory neurons were. This finding led us to posit that layer 4 of vS1 functions in a 'partially-balanced' regime, where inhibitory neurons, but not excitatory neurons, are balanced..”

In addition we now write in the discussion:
“These findings, along with the statistics of firing rates of layer 4 neurons, suggest that...”

Finally, we emphasized in the methods the mathematical definition of a partially balanced state.

- Diving further into the differences between the balanced state of inhibitory and excitatory neurons could be interesting but would likely require a dedicated study based on more than 13 neurons.

We understand the reviewer's concern regarding the sample size limitation of the inhibitory neurons in our study. While we acknowledge that a dataset of 13 neurons might seem limited, we would like to provide further context and insight into our approach:

Leveraging larger spiking dataset: Identifying inhibitory neurons in recordings in vivo is difficult, which is why the dataset of inhibitory cells is limited in size. However, we want to highlight that, beyond the set of 13 inhibitory neurons, we have access to a considerably larger spiking dataset (Fig2B). This larger dataset has enabled us to apply constraints to our computational model, enhancing the robustness of our findings.

Consistency with partially-balanced hypothesis: The hypothesis we put forward, indicating that vS1 operates within a partially-balanced regime, is substantiated by the broader context of our data. Specifically, the observation that excitatory neurons are relatively quiescent while inhibitory neurons exhibit firing rates of approximately 10 Hz (Fig2B) aligns with the expectations of such a regime. This consistency between our hypothesis and the empirical observations bolsters the credibility of our claims.

Analytical predictions and verification: Importantly, we have not solely relied on the limited dataset of 13 neurons. Our analysis extended to the analytical calculation and numerical simulations of voltage statistics of neurons in spiking neural networks. These calculations and simulations highlighted the existence of a distinct correlation structure between the distance to

spike threshold and the voltage standard deviation in inhibitory and excitatory neurons, contingent upon the average voltage standard deviation (Fig 1G, Fig 2G, Fig.5J). The analytical prediction and the behavior of the inhibitory neurons in our model is consistent with the dataset of the 13 neurons.

Thus, even with a relatively modest number of inhibitory neurons, which are hard to get in blind patch experiments in behaving animals, we believe that the broader context of the spiking data, the voltage dynamics of the excitatory neurons and the alignment with theoretical predictions, and contextual consistency with the partially-balanced hypothesis strengthen the credibility of our findings. In addition, our results are consistent with a recent study on the distance to spike threshold of vS1 neurons (Kirtani et al 2023).

We now comment on that issue at the limitation section of the discussion, where we now write:

“...we note that although the sample size of the intra-cellular recordings of inhibitory neurons in vS1 was not large (13 neurons), the alignment between the firing rates statistics of excitatory and inhibitory neurons in vS1 and the partially-balanced hypothesis, along with analytical predictions and simulations of the sub-threshold statistics of excitatory and inhibitory neurons in vS1 (Fig 2G, Fig 5J), and together with recent evidence on the differences in the distance to threshold of inhibitory and excitatory neurons in vS1 (Kirtani et al 2023) further support our claims and compensates for the challenges of obtaining inhibitory neurons in blind patch experiments in behaving animals..”

- The experimental literature related to the dynamics of vS1 should be discussed.

In terms of experimental work, to our knowledge, membrane potential recordings in vS1 from un-anesthetized mice came mostly from several papers, which we already referred to in the paper. For example, recordings without an operant behavior (Crochet and Petersen, 2006 NN, Crochet and Petersen 2011 Neuron), with an operant behavior (Sachidhanandam and Petersen 2013 NN), and with a whisker-detection operant task (Yang and O'Connor 2016 NN and Yu et al 2016, 2019).

Following the reviewer comment, we added a short discussion point on the dynamics of vS1 neurons and how it varies with whisking and during active touch, and also answered another question of the reviewer regarding richer stimuli (see below). In addition, we now cite another work that is consistent with our analysis of vS1 subthreshold activity (Kirtani et al 2023)

- A direct comparison between data from ALM and vS1 is difficult but it would be very helpful to the reader to add a bit more detail about the experiments for example in the beginning of the results section and fig 1A (for instance a description of the 'instruction cue' and recording technique). A description of the instruction cue is also missing from the methods.

We added a description of the task and a short description of the instruction cue at the beginning of the results section, the first paragraph on the analysis of vS1 neurons and in Fig1A and Fig2A. We also emphasized which recording technique was used in Fig1 and Fig2. In addition, we added to the methods a new subsection on behavior that provides more details on the behavior and the instruction cue.

- Please comment on the influence of the used recording techniques. Silicon probes will have a bias towards neurons with at least some activity whereas juxtacellular recordings will be less biased. Could that have an influence on the findings?

The reviewer is correct that silicon probes are biased (see SupFigS1C, main text and other papers in the literature, including Yu et al. 2016). For example, as we write in the main text, ALM firing rate is ~4Hz when measured using whole-cell recordings and 6Hz using silicon probes. While to exemplify the problem of narrow voltage distribution in large networks of point neurons we used the lower rates of ALM neurons, as measured using whole-cell recordings (~4Hz), following the reviewer question we also simulated networks with a wider range of firing rates values (Fig 3H of the new version):

Answer to Reviewer Figure 1

Indeed, even with lower firing rates in our models, we consistently observe smaller voltage heterogeneity compared to experimental data, emphasizing the robustness of our findings. Specifically, although the width of the voltage distribution mildly increases at lower rates (of less than 2 Hz, which is much lower than our estimate based on whole-cell recordings), the distribution is still not large due to the $1/\sqrt{K}$ factor that reduces the level of voltage heterogeneity in large networks (see section ‘Small voltage heterogeneity in one compartment models’ in Supplementary Information).

In addition, we commented on this bias under the limitation section of the discussion, where we now write:

“Furthermore, an important aspect of our research lies in establishing self-consistency between the distribution of spiking activity within the neuronal population and the distribution of sub-threshold activity. In the experimental data, the majority of supra-threshold activity is captured through extracellular recordings. It is acknowledged that this method tends to favor the selection

of neurons with higher firing rates, introducing a potential bias. However, our simulations show that voltage heterogeneity remains considerably small in models of point neurons, even when we account for lower firing rates in our modeling (Fig3H)."

- Please comment on the influence of the task design. In the vS1 dataset the whiskers are used for identifying the position of a pole. Could a more rich stimuli where the whiskers are exploring a complex object lead to another dynamical state of vS1?

We agree with the reviewer on this point and believe that it will be valuable to look at subthreshold recordings in tasks involving richer stimuli that could potentially drive vS1 neurons in a different way, thus pushing them to a different dynamical regime. We now comment on that in the discussion:

"It is important to note that a stronger excitatory drive to vS1 excitatory neurons in layer 4 could potentially steer the network to operate in a regime similar to that of the ALM network. In fact, increased excitatory input from the thalamus during whisking may be responsible to the depolarization of vS1 excitatory neurons whisking onset (Poulet et al 2008), and it is known that active touch evokes an overall robust depolarization of excitatory neurons (Crochet et al 2011, O'Connor et al. 2010). Thus, within the context of a decision-making task, it becomes conceivable that if the whiskers were engaged in exploring a more intricate object (e.g. Isett 2018), the feedforward input targeting the excitatory layer 4 neurons would increase. This increase could push the excitatory neurons closer to their spike threshold, mirroring the behavior observed in ALM neurons. It is thus possible that when confronted with more intricate and richer stimuli, the operational dynamics of the vS1 layer 4 circuit could actively adapt and transition into a balanced regime."

- 'closer to the periphery' (L68) could include cortical neurons only a few synapses away from motor neurons which, I suppose, is not what is meant in this context.

To negate this confusion we now write:

"Our work suggests that during decision making cortical excitatory neurons in areas that are at the sensory periphery are not balanced and they fire due to strong and correlated external drive, whereas cortical inhibitory neurons, and neurons in higher order areas, hover closer to their spike thresholds..."

The authors should consider to leave out the analysis of vS1. In my opinion the paper would be stronger with a more clear focus on how standard models fail to account for certain aspects (diversity in sub-threshold voltages) of data otherwise found to be in a fluctuation-driven regime and on how adding dendritic conductances can remedy this.

We agree that the paper has two main results. First, it highlights the presence of diversity in sub-threshold voltages within the dataset, shedding light on the role of dendritic conductances in cultivating such diversity within fluctuation-driven models. Second, it delves into the distinctions observed in the operational regime of ALM and vS1. Specifically, it proposes that a partially-balanced regime could elucidate the data from vS1.

While conventionally, some papers lean towards a concentrated exploration of a single result to establish a 'crystal-clear focus,' we firmly believe that from a scientific standpoint, it is imperative for the broader community to become acquainted with the nuances in subthreshold activity exhibited by neurons in vS1 and ALM. Our data suggests that these areas operate in different operating regimes, even though the task is very similar. This goes against theoretical papers that argue that the cortex operates in one regime or another. We thus prefer to maintain the analysis on vS1 within the manuscript.

Reviewer #2 (Remarks to the Author):

Neurons emit action potentials with irregular patterns, which has significant implications for the coding schemes neural circuits use to process information. Understanding the source of this variability has been a key target of neuroscience research for decades. One explanation for this variability, which is particularly relevant to cortical networks, is based on theoretical studies that show how recurrent networks of excitatory and inhibitory neurons can reach a balanced regime. In this regime, the excitatory and inhibitory drives match, and action potentials are driven by fluctuations around a mean value. This mechanism elegantly explains how the observed variability can arise in deterministic networks.

In the current manuscript, Amsalem et al. analyzed electrophysiological recordings obtained from the sensory and motor cortices of mice during behavior. The researchers first characterized the spike trains of these neurons and demonstrated their high variability over a wide range of firing rates. They then attempted to match the balanced-network model to the data and found a mismatch. The effective conductance of each neuron increased with the number of incoming synaptic inputs, which affected its integrative properties. Consequently, the neuron's membrane potential was clamped to a value determined by the balance between the excitatory and inhibitory drives, and the amplitude of the fluctuation could not be large enough to drive the spikes while providing the required variability. The authors hypothesized that placing synapses on the dendrites rather than on the soma could partially decouple the contribution of the synapse to the membrane potential and the effective conductance. This would enable a better fit between the model and the data. The researchers demonstrated this phenomenon for a single neuron using a full compartmental model and subsequently with a network model having phenomenological synapses that enabled the decoupling. Lastly, they explained the difference in behavior between sensory neurons in layer 4 and other neurons by differences in the external synaptic drive (i.e., the non-recurrent one).

This study is a beautiful work that uses rare high-quality data to study a long-standing theoretical question. The simulations and analytical work are done to the highest-level paying attention to small but important details.

We thank the reviewer for their enthusiasm on the contribution of our paper.

Major comments:

1. Re: "inevitable" in line 222 - A major factor in the discussion around figure 3 is the vanishing second term of equation 1 due to the reduction in the membrane time constant resulting from

bombardment of the neuron by synaptic inputs (see also Rapp Segev and Koch 96). This has a very clear testable prediction of the apparent time constant of the recorded cells presented in figure 1 and 2. The authors are likely to have experimentally recorded the neurons' responses to brief current pulses or long current pulses (standard procedure). They thus could fit them with multiexponential functions to estimate the effective time constant of the membrane (as done previously by the first author). It is critical to see if indeed the distribution of the experimentally derived time constant is as predicted by the model assumptions.

The first term of equation 1 (and not the second) leads to narrow voltage distributions: We thank the reviewer for pointing out this issue. We now realize that the sentence we wrote was misleading. The major factor that limits the level of voltage heterogeneity in the high-conductance regime is, in fact, **the first term** of Equation 1, and not the second term. If the rate is not high, then what determines the voltage distribution is the passive integration between the spikes, which is determined by the first term. This term is not so different between neurons when the cell is bombarded by synaptic inputs (see subsection 'Small voltage heterogeneity in one compartment models', in the SI). It results from two terms. As we wrote:

"As the total number of synaptic inputs increase, the ratio of current to conductance tends to stabilize and becomes similar across neuron."

In other words, the currents in the models are large and the time constant is small, such that their ratio becomes constant. Indeed, in the Supplementary Information we showed that in large networks the passive voltage, which is multiplication of the two terms (τ_{pass} and μ , see Eq 26 and Eq 27), converges to a number that depends only on the average rates and synaptic strength of the networks, and are thus almost the same across neurons (see also Eq 33).

In fact, as can be depicted from Figure 3G, the second term actually **decreases** the level of voltage heterogeneity. This is because the multiplication of the timescale with the firing rate of the neuron ($\tau \cdot \nu$) in the second term is correlated with the first term, V_{pas} . As a result of the minus sign in Eq1, the second term is reducing the heterogeneity level, instead of increasing it.

To negate this confusion in the main text, we removed the term 'high-conductance' and write: *"...However, this resetting effect decreases the level of voltage heterogeneity, as it pushes the voltage of neurons that fire at high rates back to the reset value. In other words, the negative sign before the $\tau_{pas,i}(V_{th}-V_r)$ term implies that this term actually decreases the level of voltage heterogeneity (Figure 3G, diamonds vs circles)."*

The distribution of the first term (and hence of the voltage distribution) increases in the extended-like model: In our extended-like model, we observed an increase in the distribution of the first term, leading to a corresponding rise in voltage heterogeneity. This phenomenon arises from the incorporation of passive dendritic effects, which diversifies the first term in Equation 1. Two factors contribute to this: first, the balancing of currents, allowing them to approach threshold levels; second, the total conductance, which can be substantial (leading to a small time constant), but not as large as in point neurons. Together, the ratio of the total current and conductance does not have to converge to the same value for all neurons.

The neuronal time constant in the data: While in the extended models the timescales are expected to be larger than their values for point neurons, it is challenging to predict specific values for total conductance and time constants in the data, as they also depend on balancing current values, a metric we lack.

Although we cannot predict specific values of time constants in the data, we followed the reviewer question regarding the neuronal timescale and tested that their values for ALM neurons are on the order of their values in the extended models. As mentioned by the reviewer, estimating the timescale can be done using current injections. However, we note that to accurately measure the time constant it is important that a short current injection be used. This is to avoid interference of active conductances (Ito & Oshima 1965, Koch, Rapp & Segev 1996). Unfortunately, the shortest current injection in the data that was used was of 200 ms. This unavoidably results with opening and closing of voltage dependent channels which will mask the "real" time constant of the cells. Even though we followed the peeling process (and developed a GUI to do so, see https://github.com/orena1/exponent_peeling), as the reviewer suggested, we found that the slow time constant (assumed to be τ_0) can sometime reach 70ms (42 ± 16 ms), which is considerably slow in respect to literature (Bindman, Meyer & Prince 1988). After subtracting the reconstructed exponent of the slowest time constant, we were able to also peel the second time constant. The mean and std of τ_1 was 4.5 ± 3 . This estimate is larger than what is known in the literature for τ_1 . Thus, we suggest that the slow time constant actually represents active conductances in the cell and the faster time constant, at least partly, represents the actual time-constant of the cell

Answer to Reviewer Figure 2

Although we can not be sure of that, if this is the case, then it means that the time-constant of the ALM neurons is $\sim 4.5\text{ms}$, which is very similar to the time constant that we used in our simulation when calculating ρ and α ($R_m = 3500\text{ohmcm}^2$ - time-constant of $\sim 3.5\text{ms}$).

2. It has been suggested that increase in average conductance by synaptic inputs is compensated by decrease in other currents such as I_h (see Waters and Helmchen 2006) or by deactivating potassium current I_A (Hoffman et al. 1997). Such mechanisms can be implemented in the integrate and fire neurons and might provide a solution that does not require the suggested dendritic mechanisms. It is likely that there is some variance in the expression of these current (I_h is particularly easy to spot). Is there any correlation between the CV of the neuron and the expression of I_h/I_A ?

Estimation of I_h currents: Following the reviewer comment, we looked at the data again. We calculated the CV of the neurons in the whole-cell recordings (Figure S1C in the revision, see also answer below). However, we have not observed any sag in any of the voltage deflection (for estimating I_h/I_A currents). We suspect that this is due to the relatively short length of the steady-state current injections (200ms) that prevents from estimating the sag. Unfortunately, it seems that the current injections are too long for clean extraction of time constants and too short for estimation of I_h using the sag of the voltage decay.

The effect of I_h conductances: Although we did not observe I_h currents in the data, to respond to the reviewer's comment we added I_h currents to both the multi-compartment neuron simulations and the network simulations. I_h channel was simulated according to its values in (Hay et al 2011), with different levels of conductances. The results of the multi-compartment neuron are depicted below:

Answer to Reviewer Figure 3

As presented, in physiological I_h conductance (0.002 S/cm²), I_h changed the results only in the case of a low number of synapses (orange lines). At much higher number of conductance the voltage is increased to a level where the I_h channels are closed (higher than -45mV) When the resting state is so close to the threshold, as is the case for ALM neurons, the I_h channels are already closed (Felton et al. 2020). Thus, adding I_h currents for neurons that are already close to their threshold only mildly change the sub-threshold statistics (at least the distance to threshold, the heterogeneity and the voltage fluctuations). In addition, we got the same results when adding I_h currents to simulations of a network of spiking point neurons (using Wang-Buzaki neurons; not shown).

In fact, we observed large effects of I_h currents on the subthreshold statistics only when we simulated extremely strong currents (green lines in Answer to Rev.Fig3). These currents were so strong that they were beyond their biological values (Hay et al 2011). In addition, the effect we observed was in the opposite direction: I_h currents seemed to reduce the level of heterogeneity.

At this high level of conductance the I_h current dominates and opposes any other currents, and the voltage distribution is very low.

To conclude, we do not know how abundant I_h currents are in our neurons, but our multicompartment and network simulations indicate that they have little effect on increasing the level of voltage heterogeneity. These nonlinear currents may have an effect at onset of inputs for neurons where their voltages are far from their spike threshold, and can also have an effect by destabilizing the asynchronous state. However, these questions are beyond the scope of our study.

3. **There seem to be one caveat** in the current story. Moving the synapses away from the soma indeed should decouple their effect on conductance from the contribution of current. Moreover, while the analysis of the visibility of the synapse is done for each synapse individually, in fact it should be done when the conductance of the membrane of the tree is set to include the average conductance of the background activity. The consequences of these two factors should be that fluctuations at the soma are mostly driven by proximal synapses. Therefore, isn't the fact that the variability only slightly dependent on the synapses number in the dendritic case (e.g., Figure 4D) reflects that the fluctuations are driven by the relatively fixed number of proximal synapses rather than what you suggest?

We will address the reviewer's comments separately.

Regarding the first comment, we appreciate the reviewer's observation that conducting the analysis with the membrane conductance set to include average background activity is essential. We want to clarify that this is indeed the approach we followed. In our Methods section, we explicitly stated, "Input resistances differ between in vitro to in vivo conditions, presumably due to the constant synaptic load which is much lower in a slice. **For estimating α , Γ we simulated the model at in vivo conditions (see below)...**". Therefore, we took into account the background activity in our simulations as suggested by the reviewer. To negate this confusion we now write in the main text:

...These simulations were conducted under 'in vivo conditions', assuming that the effect of a single synapse should be measured while others are active. This method is detailed in the Methods section "

Regarding the second comment, the reviewer suggests that the limited dependence of variability on the number of synapses in the dendritic case may indicate that fluctuations are primarily driven by a fixed number of proximal synapses. We would like to emphasize that our theoretical framework demonstrates that if the synaptic effect on somatic conductance decays faster than it affects changes in somatic current (as demonstrated in our simulations and in cable theory), then voltage heterogeneity and fluctuations can arise (see 'Finite voltage heterogeneity in extended-like point neurons' in Supplementary Information). Thus, we expect that the distant synapses contribute to the subthreshold activity.

To directly address the reviewer's concern and demonstrate that sub-threshold statistics are not solely a result of proximal synapse activity, we conducted additional simulations. Specifically, we

ran our multicompartment model with the exclusion of all non-proximal synapses, while keeping the same SD proximal synapses active (green curves in 'Answer to Reviewer Figure 4'). The resulting figure indicates that voltage statistics cannot be exclusively attributed to the activity of proximal synapses.

Specifically, we approached the comparison of different synaptic distributions in two distinct ways:

1. We created two cells, one with proximal synapses only, and the other with a full distribution, while adjusting the density of each neuron so that the total number of synapses was identical between the cells (i.e., same total number of synapses but different distribution). This is depicted in Answer to Reviewer Figure 4 A-C.
2. We created a cell with a specific number of synapses, subsequently removed the distal synapses, and compared between cells (noting that the original cell had a larger number of synapses, but the density stayed the same). This is illustrated in Answer to Reviewer Figure 4 D-F.

Answer to Reviewer Figure 4

In Figures 4A-C, it is evident that when we maintain the number of synapses constant and distribute them differently, the voltage statistics vary with the way the synapses are distributed. Across all metrics, when considering only the proximal synapse (green lines) the subthreshold statistic falls between the scenario where all synapses are located on the soma (orange lines) and the one where the synapses are evenly distributed across the cell (blue lines).

In addition, upon removing the distal synapses, it becomes apparent that they do influence the cell's behavior. They reduce the voltage SD and affect the average distance to threshold. This is depicted in Fig4D,F, where the voltage SD is larger without the distal synapses, while the average voltage is lower. These figures provide clear evidence for the impact of the distal synapses on the subthreshold statistics. The changes in voltage heterogeneity is small (Fig4E). This could be attributed to the fact that when these distant synapses are reintroduced to the complete model, their contribution to the conductance is less significant than to the current, thus preserving the variance in average voltage. If these synapses were reintegrated into the cell but located on the soma, it would result in a reduction of voltage heterogeneity (The synaptic density on the x-axis corresponds to a range of 800 to 8200 synapses in the complete model).

In the main text we now write:

“This weak dependency was not merely a result of adding distal synapses that do not affect the somatic voltage, as the distal synapses contributed both to the sub-threshold statistics. For example, with synaptic density of 0.6 synapses per microns, taking off all distance synapses (above 100 μ m) increased the SD of the voltage of the neurons from 2.6mV to 3.3mV.”

To conclude, we had estimated the parameters alpha and gamma while the conductance of the membrane of the tree was set to include the average conductance of the background activity. In addition, distal synapses do contribute to the subthreshold statistics, probably by affecting the somatic currents.

4. NMDA synapses: Another source that can contribute to the decoupling of the synapses contribution to current and conductance is non-linear currents such as the contribution of NMDA receptors. These currents are slow and can provide a lot of charge with smaller contribution to the conductance (see for example Cook and Magee 1997 & 1999 and many others), even without considering dendritic NMDA spikes. Therefore, it is possible that the resting membrane potential is boosted by non-linear currents, requiring much less conductance and hence less restrictions on the fluctuations. While NMDA currents are mentioned in the discussion, the mechanism by which they can affect the results is only poorly discussed. Please improve.

We acknowledge the reviewer's emphasis on the significance of NMDAr in diversifying subthreshold statistics. However, we were unable to locate the specific references 'Cook and Magee 97 and 99'. In Cook and Magee (2000), the authors demonstrate that the increase in dendritic EPSP amplitude with distance from the soma counteracts the filtering effects of the dendrites. This phenomenon, although relevant to the attenuation parameter (alpha) in our model, does not definitively establish whether it is exclusive to NMDA currents.

It is possible that the reviewer posits that NMDAr currents may not exert pronounced changes in somatic conductances, thereby influencing the diversification of distances to thresholds. We, however, are not sure that this is the exclusive mechanism through which NMDAr currents can help in diversifying the subthreshold activity. Our understanding lies in the non-linear effects, where an identical number of synapses can elicit varying voltage deflections, depending on the regenerative properties of NMDA receptors and their exact location.

To investigate whether NMDAr indeed contributes to diversifying subthreshold activity across different neurons, we conducted additional simulations introducing NMDA synapses (Markram et al., 2015). Preliminary results suggest that the inclusion of NMDA synapses halted the reduction in voltage heterogeneity and standard deviation of voltage as a function of the number of synapses (refer to 'Answer to Reviewer Fig 5' below). In fact, both measures actually increased with the number of synapses. It's worth noting that this observation may be contingent on the specific parameters of the NMDA synapses. We leave this to in-depth future modeling study.

Considering the reviewer's comment, and our preliminary results, we have now provided an expanded explanation of the mechanism by which NMDA receptors can influence the results in our discussion section:

“Specifically, the number of synapses required to surpass a given distance-to-spike threshold can exhibit significant variation due to non-linear NMDA currents. Once a sufficient number of synapses are activated, a substantial voltage surge can occur, even with a relatively small synapse count, owing to the regenerative properties of NMDA receptors (Major et al 2013, Palmer et al 2014). This non-linear phenomenon can diversify the range of the distances from spike thresholds. Indeed, our preliminary results indicate that incorporating NMDA receptors in multi-compartment models resulted in an elevated level of voltage heterogeneity, which notably increased with the number of synapses (not shown).”

Answer to Reviewer Figure 5

Minor comments:

1. Lines 74-78 and figure 1B it is not clear if all of these recordings are from intracellular recordings or a mixture of extra and intracellular recordings. Please specify. If there is a mixture it would be good to present the distributions separately (at least in a supp figure) because the level of confidence in spike sorting is different. Figure A suggests it is all intracellular but the numbers seem quite high for that (so the figure is a bit misleading).

The type of recordings for each subfigure was mentioned in the caption of figure 1 and figure. In addition, a comparison of the distribution of firing rates based on either whole-cell recordings or silicon probes was briefly discussed in the main text and was depicted in supplementary figure S1c.

However, given the reviewer's comment, and to minimize any confusion, in the new version of the manuscript we clarified this point by changing the cartoons in Figure 1A, Figure 1D and Figure 2B.

In addition, we emphasized the difference in the caption, and referred also in the caption of Figure 1 to Figure S1C.

2. Figure 1B it would be good to show representing example traces of cells with different CV (say 0.5, 1, 2).

We now plot 3 example traces of cells with different values of CV

3. Figure 1C – What do you make of the log-normal distribution of firing rate? Is it just an observation or does it indicate something? Is it important for your future modeling?

The log-normal distribution and the high CV are two hallmarks of the fluctuation-driven regime (see Roxin et al. 2011 and high CV see ,e.g., Brunel 2000). Specifically, it was theoretically shown in previous papers that when the neuron is fluctuation-driven its transfer function is given by an expensive non-linearity (e.g. Hansel and van Vreeswijk 2002). This property, together with a Gaussian distribution of biases that is expected in networks operating in the fluctuation-driven regime, leads to log-normal distribution of rates (Roxin et al 2011). To emphasize this point we now write:

“The log-normal distribution of firing rates and the large CV_ISI are in line with theoretical predictions based on the fluctuation-driven regime (e.g. see van Vreeswijk and Sompolinsky 1996, Brunel 2000, Roxin et al 2011).”

And also added later in the text:
“As predicted by theoretical works, the distributions of firing rates for both excitatory and inhibitory populations were well approximated by a log-normal distribution (Fig.3B-C; [19, 45])...”

4. Line 98 – Are the statistics of the ISI distributions consistent with those shown in Figure 1B? This is critical!

Following the reviewer comment we estimated the ISI distributions and calculated the CVISI in the whole-cell data. This is now plotted in supplementary figure 1C together with the CVISI of the silicon probes data for sample and delay periods. The distributions are similar and the average CVISIs are close to one in both data sets.

We do note that the CVISI for the delay period is close, but a bit smaller than 1 (~0.8), yet we get similar numbers in our model (see Fig5B blue circles). It is also worth mentioning that in the whole-cell data we have much less trials, and as a result fewer ISIs than we have in the silicon probe data. We found that this led to a small underestimation bias of the average CVISI.

5. Line 121 – This sentence does not make sense.

We rephased this paragraph to emphasize why we focused on non-whisking periods. We now write:

“We sought to compare the neural activity between vS1 and ALM. In the delayed-response task we focused on the delay period, in which the mouse movement, like licking, was minimal. Conversely, in the Go/NoGo task, no delay period existed, and it's noteworthy that vS1 activity is

influenced by whisking movements (Curtis et al 2009), particularly when the whisker contacts an object (Oconner et al 2010, Crochet et al 2011). To mitigate the impact of whisking-related activity during the task, we analyzed vS1 neuron activity during the sampling behavior phase, concentrating on non-whisking intervals (refer to Fig. S1D-F for a comparison of voltage statistics between whisking and non-whisking periods)."

6. Line 127-128 Fig 2E does not show difference between L4 and L5. And the whole discussion in this paragraph is very confusing because you compare at least four groups of neurons (L4E, L4FS L5E (of vS1) and then also ALM (which layer?), sometimes doing different comparisons in the same sentence. Please re-write this paragraph.

Following the reviewer comment, we rewrote this section. We now separate the comparisons between the different populations in vS1 and the comparison of vS1 neurons with ALM neurons.

7. The use of $V_{_}$ in Equation 1 notation is a strange choice. Why define a variable with a strange notation that you have to explain when you can use a standard notation of (V_{th} V_r) which makes much more sense.

Agree. Fixed

8. The title: "Increasing sub-threshold heterogeneity in the network by including the morphology of the neuron" could be improved (e.g. neuron's morphology)

Agree. Fixed

9. The match between figure 1C and figure 5C is pretty poor, can you comment on that?

The match for the excitatory neurons is not bad and the distributions in 1C and 5C are close to log-normal, as the theory predicts (see again Roxin et al 2011). The poor match that the reviewer mentioned is for the rate of the inhibitory neurons. In the data plotted in Fig1C, the inhibitory neurons fire at higher rates (~20Hz). However, we found that this number varies between datasets (although it was always bigger than the average rate of the putative excitatory neurons). For example, the average rate of the FS neurons in Finkelstein et al. 2021 and Li et al. 2017 was ~10Hz, closer to what we see in our model (Fig 5C). One possible reason for these differences in data sets is that in our dataset we recorded from deeper layers with respect to previous datasets, and the firing rate is higher for deeper layers (see figure below).

Answer to Reviewer Figure 6

Reviewer #3 (Remarks to the Author):

Summary:

The manuscript of Amsalem et al. investigates spiking and voltage sub-threshold neuronal dynamics, in the sensory and frontal cortex, in mice performing decision-making tasks and their underlying network mechanisms in a novel theoretical framework of 'extended-like' point neural networks. Few experimental studies have characterized sub-threshold voltage statistics in behaving animals. Understanding these statistics is fundamental to acquiring mechanistic insight into the neural mechanisms underlying cortical activity. It is also critical for advancing hypotheses on the operating regime of cortical networks.

Firstly and consistent with previous experimental studies, the authors report that neurons in their data sets exhibit Poisson-like, temporally irregular spiking activity, with rate distributions well-approximated by log-normal distributions. Secondly, the authors show that sub-threshold neural activity in the anterior lateral motor cortex (ALM) and the vibrissal somatosensory area (vS1) share common features. In particular, neurons exhibit high voltage variability and considerable mean voltage heterogeneity, with some neurons, on average, much closer to their threshold than others.

Poisson-like cortical spiking statistics is a hallmark of strongly recurrent neural networks operating in the 'fluctuation-driven' (or 'balanced regime') in which fluctuations in membrane potentials emerge from the recurrent dynamics. Nevertheless, the authors show that the standard fluctuation-driven model fails to capture the heterogeneity in sub-threshold activity in the data. They, therefore, developed a novel phenomenological network model of extended-like point neurons with synapses that mimic dendritic integration. Their work challenges classical network models by introducing an essential element of neuronal biophysics that is too often overlooked. It

is thus a significant step forward in understanding how intrinsic neuronal morphology and synaptic integration can shape population dynamics in sensory and memory systems.

The authors show how extended-like point neural networks operating in the fluctuation-driven regime allow for heterogeneous sub-threshold neural activity to emerge. Moreover, they provide a rigorous mean-field theoretical description of how extended-like features - mainly, attenuation that captures the decay in current change with the distance to the soma and visibility that captures the change in somatic conductance as a result of synapse opening - shape the network dynamics and the individual neuronal properties.

Altogether, their theoretical work points to ALM operating in a fluctuation-driven regime. In contrast, the model suggests that excitatory neurons in layer 4 of the barrel cortex are not fluctuation-driven but are driven by occasional synchronous inputs.

General comments:

The manuscript is essential to the literature on cortical neuronal dynamics and their operating regime. It is a fine example of how a theoretical approach can gain from including more biological realism in its carving and reciprocally. By varying the parameters of their extended synapses, the model can infer the voltage and spiking statistics of different cell types across different cortical layers or areas based on morphology. Even if the proposed 'extended-like' framework is not biological per se but is an elegant and powerful means to uncover how neuronal morphologies can impact network dynamics.

We thank the reviewer for their enthusiasm on the contribution of our paper.

The following points are left to the appreciation of the authors.

The manuscript emphasizes the theoretical aspects of the model (i.e., extended-like modeling framework) before its functional applications (i.e., emergence of sub-threshold selectivity to licking direction in the bistable network). An essential novelty of this model is the ability to exhibit double well attractor dynamics with extended-like neurons. It is not the primary motivation of the study that is already substantial and fine as is. However, the authors may explore how sub-threshold voltage heterogeneity (e.g., by varying the visibility parameter) might affect network performance in a possible revision. If sub-threshold heterogeneity links to behavior in the model, it could be tested in the incorrect trials in the data, thus giving the model predictive power.

As the reviewer commented, the focus of the paper is on testing the hypothesis of a fluctuation-driven regime from the point of view of sub-threshold statistics. Furthermore, we touch upon the functional implications of being in one regime or another, including the observed attractor dynamics in ALM (Inagaki et al., 2019). While we provide a novel model of extended-like neurons that exhibits a double-well dynamics, our goal in providing this model was to demonstrate that our ALM-like network operating in the fluctuation-driven regime can also account for the functional properties of ALM network. Although studying the bistable model is interesting by itself, we did not intend to provide an in-depth analysis of this model in the manuscript, as such an analysis would divert the focus from the comparison with vS1 data and the fluctuation-driven regime.

Therefore, we prefer to leave an in-depth analysis of this model and a comprehensive study of error trials in the context of bistable models for future studies.

Nevertheless, following the reviewer's suggestion, we conducted additional simulations. In the revised version of Supplementary Figure 6, we followed the reviewer's suggestion and explored how the visibility parameter impacts sub-threshold heterogeneity and network performance. Simulating the same network with different initial conditions revealed that in some cases there are transitions between attractors during the delay period, leading to error trials due to fast fluctuations and quenched disorder in the network (see also Leibovich et al., 2019). Notably, we found that a lower visibility parameter, reflecting a scenario with more 'current-based synapses,' increased neuron selectivity by allowing neurons a greater dynamic range of time-averaged voltage (increased voltage heterogeneity). This enhanced stability of attractors with dendritic inclusion led to a decrease in error trials and an overall increase in network performance. These findings have been incorporated into the revised Supplementary Figure 6 and are depicted below, both for the network consisting of spiny stellate cells and the network consisting of layer 5 pyramidal cells.

Answer to Reviewer Figure 7

Given this result, we have taken the reviewer's advice to emphasize the functional implications of dendritic inclusion. We have reworked the concluding section of the manuscript, titled 'Networks of extended-like neurons that operate in a fluctuation-driven regime can account for the supra- and sub-threshold statistics of ALM neurons,' to underscore the significance of our results in relation to the role of dendrites in neural function. In addition, we added the following sentence in the discussion: "Notably, our inclusion of dendrites in the extended model seems to enhance the stability of memory states".

Finally, and on the same line of thought, reference [34] contains data with optogenetic perturbations of ALM. Optogenetic perturbations could switch ALM from operating in a fluctuation-driven regime to a partially balanced state. The author could choose to analyze and show the level of sub-threshold heterogeneity in perturbed trials and reassert the potency of the model by showing similar effects in their simulations.

We rephrase the reviewer's suggestion as follows: When we activate the inhibitory neurons, a paradoxical effect might occur, leading to a potential reduction in the firing rates of both inhibitory and excitatory neurons. This reduction could continue until the excitatory neurons become completely inactive. At this juncture, the ALM could conceivably operate in a partially balanced state, akin to the vS1 layer 4 network. Unfortunately, the analysis recommended by the reviewer necessitates data from whole-cell recordings of pyramidal cells during optogenetic perturbations, a dataset that we lack for behavior-related activities. Our current dataset only comprises perturbation data obtained via silicon probes (above threshold). Though we do possess whole-cell data involving perturbations outside the context of behavior (Extended Data Figure 5 in reference [34]), this dataset is confined to a limited number of neurons.

REVIEWER COMMENTS

Reviewer #1 (Remarks to the Author):

Amsalam et al. have improved the manuscript and addressed many issues raised by the referees. However, I am still not convinced that there are any compelling reason to include the vS1 analysis in the manuscript and would suggest the authors to reconsider.

Yu et al. [38] found excitatory L4 neurons at rest to have a membrane potential far from threshold and low firing rate, whereas the inhibitory neurons show higher firing rates. Further they found the activity of the excitatory neurons to be largely driven by synchronous input. It is already clear from the original study that the excitatory neurons are not fluctuation driven.

The findings of Yu et al. are likely specific to the used stimulus. As mentioned by Ahmadian and Miller (2021), the dynamical regime of cortex depends on experimental parameters such as species, brain state, attention, arousal, stimulus drive and cortical area. Ahmadian and Miller cite a list of the experimental studies conducted the last decades which illustrates how the dynamical regime depends on experimental details. This questions the value of a comparative study of two specific datasets will be.

The authors provide a list of arguments in the rebuttal, for example 'Our data suggests that these areas operate in different operating regimes, even though the task is very similar. This goes against theoretical papers that argue that the cortex operates in one regime or another.'

It is not surprising that two cortical areas can operate in different dynamics regimes (or even two different layers in the same area). For example, a given stimulus might lead a sensory area to be in a fluctuation-driven regime, but other sensory areas will be dominated by inhibition with excitatory neurons far from threshold and with very low spike rate. I'm not aware of theoretical papers that argue that all of cortex are in one regime or another.

Further, I don't follow the authors argument that the behavioral tasks used are very similar. The authors analyze data from previously published studies and, if I understand the authors correctly, argue they are very similar because the animals are engaged in decision-making tasks (with similar timings). Firstly, a 2AFC task using auditory stimuli and a Go-Nogo task using tactile stimuli differs in important aspects. Secondly, the dynamical regime of vS1 is likely determined by the stimulus drive and whether it is part of a decision-making task is not important.

Reviewer #2 (Remarks to the Author):

The authors answered all my concerns with additional simulations, analysis and textual changes. I find the preliminary findings regarding the contribution for NMDA current interesting and I think this deserves further exploration in future studies.

I have no further comments and congratulate the authors for their achievement.

Reviewer #3 (Remarks to the Author):

Dear authors,

Thank you for providing a thoughtful and thorough response to my comments. I appreciate the effort you have put into fully addressing each concern. I am pleased to inform you that your revisions have successfully addressed my major and minor points.

The additional simulations, new figures, and discussions you have incorporated into the revised manuscript have significantly enhanced its clarity and strength. In particular, including the visibility parameter and its impact on sub-threshold heterogeneity and network performance adds depth to your findings. The simulations you conducted demonstrate the significance of dendritic inclusion, further supporting that dendritic morphology plays a crucial role in the stability of attractor dynamics and memory states.

Regarding the suggestion concerning optogenetic perturbations of ALM, I understand the limitations of your current dataset and the unavailability of whole-cell recordings during behavior-related activities. Your explanation clearly justifies not delving more deeply into such an analysis in the manuscript.

Based on your revisions and the clear and comprehensive manner in which you have addressed my comments, I believe the manuscript is now in a suitable and robust state for publication. I recommend submitting it to the editors without further delay.

Thank you again for your attention to detail and for considering my feedback. I genuinely appreciate your research and the contributions it brings to the field.

Best regards,

A.M.

Sub-threshold neuronal activity and the dynamical regime of cerebral cortex

Oren Amsalem, Hidehiko Inagaki, Jianing Yu, Karel Svoboda, and Ran Darshan

Answer to the Reviewer's comments

We thank the reviewers for their insightful and constructive reviews. We have revised our manuscript in response to their comments and addressed all the reviewer's concerns.

We restructured our manuscript in response to reviewers' 1 main concern. In addition, we provided detailed answers to various comments raised by reviewer 1. We respond in detail to individual points below. Reviewer 1 comments are set in **BLACK** and our replies in **BLUE**. In the revised manuscript, edited and newly added texts are highlighted in **BLUE**.

Reviewer #1 (Remarks to the Author):

Amsalem et al. have improved the manuscript and addressed many issues raised by the referees. However, I am still not convinced that there are any compelling reason to include the vS1 analysis in the manuscript and would suggest the authors to reconsider.

We appreciate the detailed reading and power invested in the review. It is unfortunate that we were not able to convince the reviewer of the importance of including the modeling and analysis of vS1 data. Following the strong objection of the reviewer, we have removed all relevant data, analysis, and figures regarding vS1 from the main text. As suggested in their current and previous reviews, we are now focusing the paper on the analysis of ALM activity, the theory, and the modeling part. We refer to vS1 data only in the discussion, where we discuss previous results, and there we point the reader to the supplementary information for more details.

We provide detailed answers to the reviewer's comments below. However, after carefully reading them, we would like to stress that we agree with the reviewer's point that the partial-balance regime of vS1 likely results from the stimulus drive (relative to recurrent inhibition), as we previously commented on in the manuscript's earlier version.

In fact, using both the model and literature on connectivity in layer 4 vS1, we proposed, as the reviewer pointed out, that the primary difference between ALM and the layer 4 vS1 network primarily arises from the stimulus drive rather than other experimental parameters, placing vS1 in a partially-balanced regime. Specifically, we suggested that the imbalance in the excitatory population, but not the inhibitory population, stems from the ratio of I_E/I_I relative to the ratio of g_{EI}/g_{II} (see also van Vreeswijk & Sompolinsky 98). External stimuli that alter this I_E/I_I ratio can then shift vS1 into a fully-balanced regime. This stands in contrast to Tan et al. 14, which discusses how excitatory neurons in V1 can transition into a fluctuation-driven regime without delving into the role of the inhibitory neurons in this transition. In other words, the mechanism for how the network transitions from a partially-balanced to a fully-balanced regime goes beyond

stating that there can be sensory areas in which the excitatory neurons are far from the threshold with a very low spike rate.

We previously suggested, as a discussion point, that the fact that layer 4 vS1 and ALM operate in different dynamical regimes has different functional roles (e.g., detection Vs. memory). Following the reviewer's comment, we agree that it remains an unresolved question whether the same observation on vS1 being in a partially-balanced regime would persist also under different decision-making tasks.

Yu et al. [38] found excitatory L4 neurons at rest to have a membrane potential far from threshold and low firing rate, whereas the inhibitory neurons show higher firing rates. Further they found the activity of the excitatory neurons to be largely driven by synchronous input. It is already clear from the original study that the excitatory neurons are not fluctuation driven.

We agree with the reviewer that some of the observations on layer 4 neurons in vS1 were made in Yu et al. However, these observations do not clearly indicate that excitatory neurons are not fluctuation-driven, and it is certainly not evident for inhibitory neurons.

The characterization of the network regime depends on three key factors: 1) the distance to threshold, 2) the level of voltage fluctuations, and 3) the heterogeneity across neurons. Each of these factors relies, in a self-consistent manner, on the spiking activity of both excitatory and inhibitory populations in the network. Observing neurons that are far from threshold and barely spiking is insufficient to conclude that the network is not fluctuation-driven. For example, although the neurons are far from threshold, their currents can be balanced and have large fluctuations, but still not large enough to generate sufficient firing. Furthermore, this observation fails to elucidate the dynamic state of the network, as it overlooks the crucial contribution of the inhibitory population, which plays an integral role in shaping the network dynamics. In addition, the inhibitory neurons can be far from their threshold and fire at high rates due to large synchronous fluctuations, and not because they are close to their threshold and spike as a result of the fluctuations.

This is precisely where our model provides insight. It demonstrates, quantitatively, that the spiking and subthreshold statistics of both the excitatory and inhibitory neurons in the dataset can be explained self-consistently if the network operates in a partially-balanced regime. In this regime, excitatory neurons, but not inhibitory neurons, are unbalanced and are driven by synchronous fluctuations. Furthermore, our theory and modeling account for the correlations between voltage fluctuations and the distance to the threshold, which differ between populations. It also highlights that synchronous inputs must be considered to explain the skewness of the voltage for the excitatory neurons that are far from their spike threshold.

In simpler terms, the qualitative statement that excitatory neurons are far from the threshold, have low firing rates, and are largely driven by synchronous input offers only partial insights into the dynamical regime of vS1 network.

That being said, we acknowledge that the comparison with ALM neurons has its limitations, especially as it does not involve exactly the same tasks used to estimate the network state. Consequently, and in response to the reviewer's request, we have completely removed vS1 data analysis and modeling from the main text.

The findings of Yu et al. are likely specific to the used stimulus. As mentioned by Ahmadian and Miller (2021), the dynamical regime of cortex depends on experimental parameters such as species, brain state, attention, arousal, stimulus drive and cortical area. Ahmadian and Miller cite a list of the experimental studies conducted the last decades which illustrates how the dynamical regime depends on experimental details. This questions the value of a comparative study of two specific datasets will be.

We agree with the reviewer that multiple factors contribute to the dynamical regime. This is why we thought a comparative study between the two brain regions under 'similar conditions' would be interesting. The analysis was done on a dataset collected using the same protocols (same lab) within awake mice that are engaged in a task. The sensory cue in this task relied either on whisking (vS1 and ALM data), or auditory cue (ALM data). However, at least within the ALM dataset, the analysis we did was not so different if the task was auditory or whisking. Thus, while we acknowledge that there are major differences between the tasks (see below), given all the limitations, we thought that many of the parameters between the two experiments were similar and that it was interesting to make this comparison.

In fact, using both the model and literature on connectivity in layer 4 vS1, we proposed, exactly as the reviewer pointed out, that the primary difference between the two areas arises from the stimulus drive rather than the experimental parameters mentioned earlier, placing vS1 in a partially-balanced regime (see also answer to the first comment).

We acknowledge the limitations of the comparison and comment on them in several places in the previous version, as well as in our answer to the reviewers' comments. Yet, as mentioned above, we follow the reviewer's suggestion and do not include vS1 dataset and modeling part in the main text but only mention it in the discussion.

The authors provide a list of arguments in the rebuttal, for example 'Our data suggests that these areas operate in different operating regimes, even though the task is very similar. This goes against theoretical papers that argue that the cortex operates in one regime or another.'. It is not surprising that two cortical areas can operate in different dynamics regimes (or even two different layers in the same area). For example, a given stimulus might lead a sensory area to be in a fluctuation-driven regime, but other sensory areas will be dominated by inhibition with excitatory neurons far from threshold and with very low spike rate. I'm not aware of theoretical papers that argue that all of cortex are in one regime or another.

We agree with the reviewer's observation that two cortical areas and different layers can operate in different regimes, but thought that this wasn't the prevailing view in the community. Our sincere belief was that our work could illustrate these differences, both between areas and across different

layers. Our model offers a simple mechanistic explanation for why, in one brain region, both excitatory and inhibitory populations are balanced, while in another area, only the inhibitory neurons exhibit balance.

For instance, Ahmadian and Miller initiated and concluded their discussion by arguing that the cortex as a *whole* is loosely balanced. The tightness of the balance is integral to the description of the cortical regime and is part of the ongoing debate in the field. They explicitly stated: '...we believe that at least sensory, **and perhaps all of**, the cortex operates in a regime in which the inhibition and excitation neurons received are loosely balanced.' In other words, they did not delve into specific layers or populations but referenced the cortex as a whole. Interestingly, they did postulate that there might be differences between sensory and other cortical areas, which is one of our findings. Our results not only demonstrate differences between areas but also reveal distinctions between layers and neuronal populations. Although we did not explicitly focus on the tightness of the balance, our data and modeling suggest that it is tight for ALM neurons and also for vS1 inhibitory neurons, but not for vS1 excitatory neurons.

We also agree with the reviewer that, as our model demonstrates, a given stimulus might induce layer 4 vS1 to be in a fluctuation-driven regime (see also answer to the first comment).

Again, following the reviewer's suggestion, we have removed all vS1 data and modeling from the main text.

Further, I don't follow the authors argument that the behavioral tasks used are very similar. The authors analyze data from previously published studies and, if I understand the authors correctly, argue they are very similar because the animals are engaged in decision-making tasks (with similar timings). Firstly, a 2AFC task using auditory stimuli and a Go-Nogo task using tactile stimuli differs in important aspects. Secondly, the dynamical regime of vS1 is likely determined by the stimulus drive and whether it is part of a decision-making task is not important.

We agree with the reviewer that the behavioral tasks have their differences, and we acknowledge this limitation. As noted above, the fact that it is a 'similar' task serves to minimize other differences, such as arousal and variations between species and lab protocols. We also note that we analyzed the 2AFC with a tactile stimulus that is reminiscent of the tactile stimulus in the Go/NoGo task and not only an auditory stimulus (and that the differences for ALM neurons in the context of our analysis were minimal). Still, we admit that there are important differences between 2AFC and Go/NoGo tasks.

We also agree with the reviewer's perspective that the dynamic state of vS1 is likely influenced by the stimulus drive (see above). It remains an unresolved question whether the distinct functional roles of vS1 and ALM regions in the decision-making task contribute to their operation in distinct dynamical regimes.

Given the strong opposition of the reviewer, we have removed the comparison between the two datasets from the main text.

Reviewer #2 (Remarks to the Author):

The authors answered all my concerns with additional simulations, analysis and textual changes. I find the preliminary findings regarding the contribution for NMDA current interesting and I think this deserves further exploration in future studies. I have no further comments and congratulate the authors for their achievement.

We thank the reviewer for their encouraging words, their insightful comments and the detailed reading invested in the review process.

Reviewer #3 (Remarks to the Author):

Dear authors,

Thank you for providing a thoughtful and thorough response to my comments. I appreciate the effort you have put into fully addressing each concern. I am pleased to inform you that your revisions have successfully addressed my major and minor points.

The additional simulations, new figures, and discussions you have incorporated into the revised manuscript have significantly enhanced its clarity and strength. In particular, including the visibility parameter and its impact on sub-threshold heterogeneity and network performance adds depth to your findings. The simulations you conducted demonstrate the significance of dendritic inclusion, further supporting that dendritic morphology plays a crucial role in the stability of attractor dynamics and memory states.

Regarding the suggestion concerning optogenetic perturbations of ALM, I understand the limitations of your current dataset and the unavailability of whole-cell recordings during behavior-related activities. Your explanation clearly justifies not delving more deeply into such an analysis in the manuscript.

Based on your revisions and the clear and comprehensive manner in which you have addressed my comments, I believe the manuscript is now in a suitable and robust state for publication. I recommend submitting it to the editors without further delay.

Thank you again for your attention to detail and for considering my feedback. I genuinely appreciate your research and the contributions it brings to the field.

We thank the reviewer for their encouraging words, their insightful comments and the detailed reading invested in the review process.

REVIEWERS' COMMENTS

Reviewer #1 (Remarks to the Author):

Amsalem et al. have addressed my concerns and strengthen the manuscript with a more clear focus by removing the vS1 analysis.

I thank the authors for their detailed answers to my comments.